# ReGUIDE: Data Efficient GUI Grounding via Spatial Reasoning and Search

## Abstract

Recent advances in Multimodal Large Language Models (MLLMs) have enabled autonomous agents to interact with computers via Graphical User Interfaces (GUIs), where accurately localizing the coordinates of interface elements (e.g., buttons) is often required for fine-grained actions. However, this remains significantly challenging, leading prior works to rely on large-scale web datasets to improve the grounding accuracy. In this work, we propose Reasoning Graphical User Interface Grounding for Data Efficiency (ReGUIDE), a novel and effective framework for web grounding that enables MLLMs to learn data efficiently through self-generated reasoning and spatial-aware criticism. More specifically, ReGUIDE learns to (i) self-generate a language reasoning process for the localization via online reinforcement learning, and (ii) criticize the prediction using spatial priors that enforce equivariance under input transformations. At inference time, ReGUIDE further boosts performance through a test-time scaling strategy, which combines spatial search with coordinate aggregation. Our experiments demonstrate that ReGUIDE significantly advances web grounding performance across multiple benchmarks, outperforming baselines with substantially fewer training data points (that is, only 0.2% samples compared to the best open-source baselines).

## 1 Introduction

Graphical User Interface (GUI) agents—i.e., Multimodal Large Language Model (MLLM) agents that interpret visual screen contents and generate web actions in natural language to navigate the web environment—have shown promising capabilities in web navigation tasks as MLLMs continue to improve in general decision-making ability (OpenAI, 2024; Anthropic., 2024; Qin et al., 2025; Zheng et al., 2024). Despite the recent significant efforts, however, a substantial gap remains between the performance of these agents and that of proficient human users (Chen et al., 2025), particularly in complex or long-horizon tasks (Zhang et al., 2025). This gap arises mainly from two challenges: limited visual understanding of web pages (Wang et al., 2024) and insufficient web domain-specific decision-making ability (Gou et al., 2024). To address this, prior work typically decomposes the problem into two parts: (i) building an MLLM that can interpret the visual input (Zheng et al., 2024) and (ii) employing a language model that performs decision-making and planning based on the MLLM's interpretation (Qin et al., 2025; Hong et al., 2024), where the major bottleneck lies in the visual understanding component (Gou et al., 2024).

Building an MLLM that can predict the exact pixel coordinates of target region on the screen (e.g., buttons) has shown great promise in providing the visual understanding required by another LLM for effective decision-making (Gou et al., 2024). However, grounding remains challenging as it requires fine-grained skills such as reasoning and the manifestation of spatial understanding (Ma et al., 2025; Zhao et al., 2025). Recent approaches have emphasized the importance of large-scale, high-quality image–instruction pairs for training MLLMs through supervised fine-tuning (SFT) (Lin et al., 2024; Gou et al., 2024). Gou et al. (2024) proposes synthesizing diverse referring expressions that can express the same object in several views. However, such SFT models depend on costly data curation to perform well, which poses a major scalability challenge (Huang et al., 2025).

To this end, we focus on extracting rich information from web image data to achieve data-efficient grounding by learning to explain the coordinate prediction process in natural language and by leveraging spatial priors to maximize the utility of visual input.

**Contribution.** We propose Reasoning Graphical User Interface Grounding for Data Efficiency (ReGUIDE), a novel and effective GUI coordinate grounding method for web agents. Specifically, ReGUIDE is composed of a two-stage training: (i) self-generation of image explanation through its own reasoning, and (ii) criticism of the localization prediction via spatial priors. First, ReGUIDE learns to generate the language description of the given web image that can guide itself to correctly predict the coordinate, where this prediction accuracy is used as the reward for online RL. Then, by leveraging a spatial prior—i.e., the fact that augmentations such as cropping lead to equivariant changes in the target coordinates—the model criticizes its predictions by ensuring consistent outputs under the same language description across augmented image–coordinate pairs.

Furthermore, we introduce an inference-time scaling strategy for ReGUIDE, which integrates spatial search. Specifically, the model generates multiple localization predictions and crops the region where the target is likely to exist based on the predictions. In this region, we generate multiple coordinate candidates, then aggregate the candidates into a single coordinate via statistical voting strategy (i.e., Kernel Density Estimation (Rosenblat, 1956) (KDE)-based voting).

We demonstrate the effectiveness of ReGUIDE through evaluations on multiple web-grounding datasets and agent-setting benchmarks. Notably, ReGUIDE enhances web coordinate grounding performance beyond prior methods, achieving the state-of-the-art performance in web grounding. For instance, ReGUIDE improves the grounding accuracy of Qwen-2.5-VL-3B from 55.5% to 87.3% on SCREENSPOT (CHENG ET AL., 2024A) and from 23.9% to 44.3% on SCREENSPOT-PRO (LI ET AL., 2025). Moreover, our experimental results show that ReGUIDE indeed guides the GUI agent to improve the overall decision-making ability, showing a significant performance improvement in agentic tasks. Beyond demonstrating the overall effectiveness of ReGUIDE, we further analyze the contribution of each component and show how they individually enhance grounding performance.

## 2 RELATED WORK

**Graphic User Interface (GUI) grounding.** Recent advances in pixel-level GUI grounding have demonstrated mapping natural language instructions to screen coordinates without relying on HTML or DOM structures (Shaw et al., 2023). Several prior works (Xu et al., 2024; Wu et al., 2024; Cheng et al., 2024b; Lin et al., 2024) train on over a million synthesized screenshots and achieve superior performance. Gou et al. (2024) and Yang et al. (2025b) further shows that synthesizing and augmenting text instructions can lead to further improvements. However, current approaches depend on massive annotated datasets, incurring substantial computation and labeling costs, and neither leverages the reasoning capabilities of large language models to improve localization under data-scarce or out-of-distribution conditions.

**Reinforcement learning in MLLM.** Reinforcement learning has emerged as a powerful mechanism to fine-tune multimodal large language models (MLLMs) via self-improvement and feedback. Self-Critical Sequence Training (Rennie et al., 2017) researched policy-gradient optimization for image captioning. RLHF-V (Yu et al., 2024) formulates multimodal RLHF under constrained optimization to jointly maximize helpfulness and minimize unsafe outputs. More recently, Vision-R1 (Huang et al., 2025) generates reasoning trajectories and applies iterative policy optimization to boost multimodal math reasoning. For GUI grounding, several concurrent RL-based approaches (Lu et al., 2025; Xia & Luo, 2025; Lian et al., 2025; Yuan et al., 2025; Yang et al., 2025a) optimize grounding decisions via reinforcement learning, often emphasizing dataset curation, and reward shaping. In contrast, ReGUIDE takes a fundamentally different perspective which focus on leveraging spatial priors through transformation consistency training and test-time spatial search rather than reward design. These components can be operate agnostically for existing models, ReGUIDE can be applied orthogonally to existing models and improve performance generally as shown in Table 1 and Table 2

**Test-time scaling.** Recent works explored that scaling test-time computation, such as best-of-N sampling, can be even better than scaling train-time computation for performance (Snell et al., 2024). Specifically, test-time scaling strategies improve LLM performance by generating numerous candidate outputs and selecting the best (Snell et al., 2024; Lee et al., 2025; Hosseini et al., 2024). To enhance decision-making, external verifiers are often employed to evaluate and refine these outputs (Hosseini et al., 2024). In the localization task, recent work framed the localization as a search problem and suggested a test-time scalable strategy (Wu & Xie, 2024; Luo et al., 2025). Unlike prior approaches that only aggregate predictions by selecting the most confident point without proposing any training

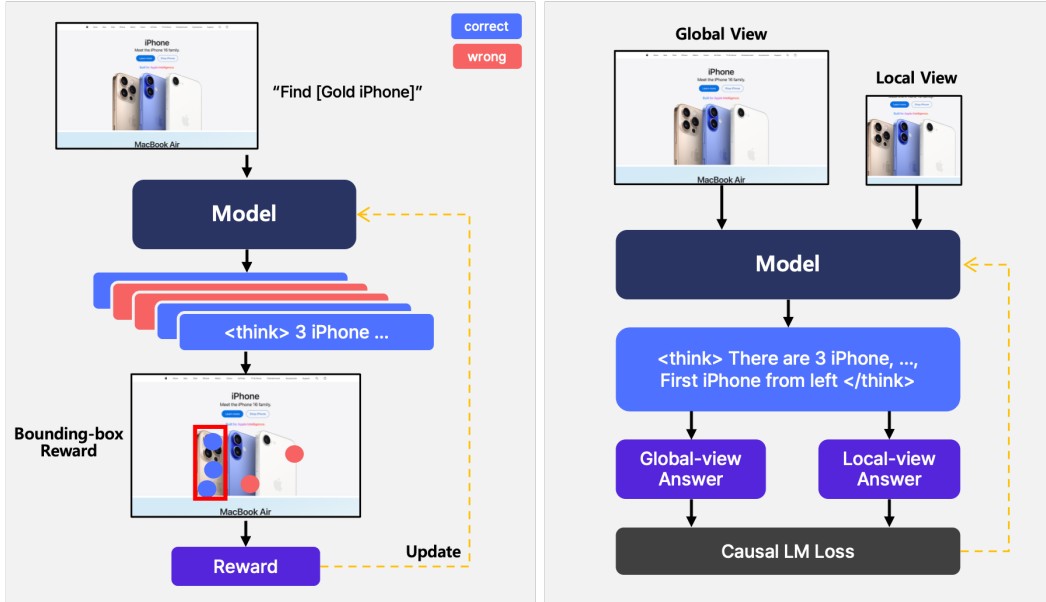

(a) Learning to explain web images via reasoning     (b) Learning to predict consistent coordinates

Figure 1: **Overview of training ReGUIDE. Left**: The model rolls out multiple (reasoning, coordinate) pairs; rewards the point that falls inside the ground-truth box. **Right**: Trains the model on paired full-image and cropped views, sharing the reasoning tokens while adjusting the coordinate, which provides multi-view consistency and improves grounding performance.

scheme, we additionally introduce a novel learning method and a more advanced aggregation strategy motivated by the Gaussian-shaped likelihood distribution of coordinate tokens observed in Figure 3. As shown in Table 11, this approach enables ReGUIDE to outperform alternative methods.

## 3   DATA EFFICIENT GUI GROUNDING VIA SPATIAL REASONING AND SEARCH

In this section, we present Reasoning Graphical User Interface Grounding for Data Efficiency (ReGUIDE), a Graphical User Interface (GUI) grounding (i.e., coordinate prediction) training framework that self-generates the language description of the image through reasoning, and criticizes the prediction with spatial priors. We present the core training method in Section 3.1, and then the test-time search method in Section 3.2. The overview of ReGUIDE is depicted in Figure 1.

**Problem setup.** We describe the problem setup of our interest, namely the GUI grounding: training an MLLM that can predict the coordinate of the interface (e.g., 'close button') in the web image based on the given instruction (e.g., 'close the window'). Formally, given an input instruction $x_{\text{inst}}$ and image $x_{\text{img}}$, the MLLM $\mathcal{M}$ generates the output which consists of reasoning path $r$ and predicted coordinate $\mathbf{c} = (c_1, c_2)$, i.e., $r, \mathbf{c} \sim \mathcal{M}(\cdot | x_{\text{inst}}, x_{\text{img}})$. Here, we denote the ground-truth region in the pixel coordinate space corresponding to $(x_{\text{inst}}, x_{\text{img}})$ as $\mathcal{X}_{\text{gt}} \subseteq [0, W] \times [0, H]$, where $W$ and $H$ are the width and height of $x_{\text{img}}$. Any coordinate $\mathbf{c} \in \mathcal{X}_{\text{gt}}$ is considered a correct prediction, as all such points result in the same GUI behavior (e.g., triggering the intended button or link).

### 3.1   REGUIDE: REASONING GRAPHICAL USER INTERFACE GROUNDING FOR DATA EFFICIENCY

We describe the core training pipeline of ReGUIDE. The key idea is to extract rich information from web images, enabling the model to generalize even with a relatively small training dataset. To this end, we train the model to self-explain the reasoning behind its predictions, as learning an explicit reasoning process—rather than merely memorizing input-output patterns—is crucial for generalization (Nye et al., 2021; Lampinen et al., 2022). Building on this learned reasoning, we

further refine the predictions through a consistency enforcement step, which assesses the consistency of the output under image augmentations while maintaining the same reasoning trace.

**Learning to explain GUI images via reasoning.** At the first stage, we train the LLM to reason about the exact coordinate of the instruction on the screen, where we employ online reinforcement learning (RL) to make MLLM self-evolve by using grounding accuracy as a reward. This enables the model to generate its own text reasoning and evolve without relying on externally provided language descriptions, thus saving the language annotation cost. RL also tends to generalize better than supervised fine-tuning (SFT), which may suffer from memorization (Nye et al., 2021).

For a given instruction $x_{\texttt{inst}}$ and an image $x_{\texttt{img}}$, we define two reward functions to train MLLM with RL. Specifically, we consider (i) the result of ground truth prediction as the primary reward and (ii) an auxiliary formatting reward that encourages the model to wrap its reasoning trace between '`<think>`' and '`</think>`' tags, ensuring that the generated reasoning follows the desired format for downstream usage by following recent work in LLM reasoning (Huang et al., 2025; Guo et al., 2025). Formally, we define the reward function $R(\cdot, \cdot)$ as follows:

$$R(r, \mathbf{c}) = \mathbf{1}[\mathbf{c} \in \mathcal{X}_{\texttt{gt}}] + \lambda \cdot \mathbf{1}[\text{format}(r) = \texttt{<think>} \cdots \texttt{</think>}], \qquad (1)$$

where $\mathbf{1}[\cdot]$ denotes the indicator function that returns one if the condition holds and zero otherwise, $\text{format}(r)$ extracts the outermost structure of the generated reasoning trace to check whether it is properly enclosed between `<think>` and `</think>`, and $\lambda \in \mathbb{R}^+$ is a hyperparameter that balances the contribution of the formatting reward, where we use $\lambda = 0.1$ throughout the experiment.

The overall RL objective is to maximize the expected reward:

$$\max_{\mathcal{M}} \, \mathbb{E}_{x_{\texttt{inst}}, x_{\texttt{img}}} \, \mathbb{E}_{(r, \mathbf{c}) \sim \mathcal{M}(\cdot | x_{\texttt{inst}}, x_{\texttt{img}})} \left[ R(r, \mathbf{c}) \right]. \qquad (2)$$

To this end, we adopt Group Relative Policy Optimization (GRPO) (DeepSeek, 2024) as our RL algorithm, as GRPO enhances both efficiency and stability by sampling multiple candidate outputs per input and computing group-relative rewards. This overall process allows the model to explore diverse reasoning strategies and progressively reinforce those that contribute to accurate grounding, without relying on explicit reasoning supervision.

**Learning to predict consistent coordinates under transformations.** While the RL stage enables the model to predict coordinates within the correct ground truth region, it does not necessarily guide the model toward predicting the most precise or stable point, typically the center of the target element (as the target UI element is annotated with a bounding box, making the center a natural target). Additionally, the model may be sensitive to variations in scale and cropping, which can degrade performance in real-world scenarios where UI elements may appear at different sizes or positions.

To address these limitations, we introduce a further training phase that enforces spatial precision and view consistency. Specifically, we continually train the MLLM to predict the center of the ground-truth region, while jointly enforcing consistent predictions under image transformations through spatial priors. In particular, we focus on constraining consistency between the model's predictions on the global view (full image) and the local view (a zoomed-in crop covering the ground-truth region). Notably, such global-local view consistency has been identified as a key factor in the success of prior self-supervised vision methods (Caron et al., 2021; Oquab et al., 2023; Hjelm et al., 2018). Furthermore, we make the local view follow the reasoning process of the global view (as it typically contains relative spatial cues), forming a self-distillation of the reasoning process.

Formally, we collect two data points, namely the global view data $\mathbf{d}_{\texttt{global}}$ and local view data $\mathbf{d}_{\texttt{local}}$, where each data point consists of image, instruction, reasoning, and target coordinates. Here, both data points consist of the same instruction and reasoning, but with different images and target coordinates (due to the spatial transformation). To ensure high-quality reasoning, we repeat the the sampling 4-time from given reasoning and just select the reasoning which consistently predict the target region in the global image $x_{\texttt{img}}$, i.e., select $r$ from $r \sim \mathcal{M}(\cdot \mid x_{\texttt{inst}}, x_{\texttt{img}})$ such that $\{\mathbf{c}_i\}_{i=0}^3 \sim \mathcal{M}(\cdot \mid r, x_{\texttt{text}}, x_{\texttt{img}}), \bigwedge_{i=0}^3 \mathbf{c}_i \in \mathcal{X}_{\texttt{gt}}$. For the global view data, we use the non-transformed image $x_{\texttt{img}}$ and the center of the ground truth region $\texttt{center}(\mathcal{X}_{\texttt{gt}})$, while for the local view data, we use a randomly cropped image $\texttt{crop}(x_{\texttt{img}})$ and the center of the transformed ground truth region, denoted as $\texttt{center}(\mathcal{X}_{\texttt{gt}}^{\texttt{crop}})$. Namely, the global and local view data are defined as:

$$\mathbf{d}_{\texttt{global}} := \big(x_{\texttt{inst}}, x_{\texttt{img}}, r, \texttt{center}(\mathcal{X}_{\texttt{gt}})\big), \quad \mathbf{d}_{\texttt{local}} := \big(x_{\texttt{inst}}, \texttt{crop}(x_{\texttt{img}}), r, \texttt{center}(\mathcal{X}_{\texttt{gt}}^{\texttt{crop}})\big).$$

For training, the model processes the original global view and the newly created local view in the same batch, and is supervised via next-token prediction loss to generate the updated coordinates:

$$\min_{\mathcal{M}} \mathcal{L}(\mathbf{d}_{\text{global}}) + \mathcal{L}(\mathbf{d}_{\text{local}}), \quad \text{where} \quad \mathcal{L}(\mathbf{d}) = -\log \mathcal{M}(r, \mathbf{c} \mid x_{\text{inst}}, x'_{\text{img}}), \qquad (3)$$

where $\mathbf{d} = (x_{\text{inst}}, x'_{\text{img}}, r, \mathbf{c})$ and $x'_{\text{img}}$ denotes the input image used in the given view (i.e., $x_{\text{img}}$ for the global view and $\text{crop}(x_{\text{img}})$ for the local view).

## 3.2 Test-time Scaling with Spatial Search and KDE-Based Aggregation

Despite their strong reasoning capabilities via natural text, LLMs inherently struggle with coordinate prediction due to their limited awareness of ordinal relationships among numeric tokens. To alleviate the issue, we propose a scalable inference method for trained models. Specifically, we introduce a two-stage inference method, namely, composed of the **(i) cropping** and the **(ii) voting** stages. Here, the key idea for each stage is to (i) zoom in on the image where the UI element is likely to exist, and (ii) predict multiple coordinates, which are finalized to a single coordinate by using Gaussian kernel density estimation. An illustration of our test-time spatial search strategy is provided in Appendix D.

**Cropping: Zooming into the UI element area.** To crop the region where the UI element is likely to exist, we first predict multiple coordinates on the full image. Then, we treat these predictions as samples from a probability distribution indicating the target's likely location. Here, we use Kernel Density Estimation (KDE) to analyze these initial samples and identify the region of highest prediction density—the area where the model is most consistently expected to be the target. Thus, we choose the highest density point as the center of the cropped Region of Interest (RoI), effectively allowing the model to "zoom in" on the most promising area.

Concretely, given an input image $x_{\text{img}}$ and instruction $x_{\text{inst}}$, we sample $N$ initial predictions: $\mathcal{C} := \{\mathbf{c}^{(i)}\}_{i=1}^{N}$, where $(r^{(i)}, \mathbf{c}^{(i)}) \sim \mathcal{M}(\cdot \mid x_{\text{inst}}, x_{\text{img}})$. Then, we apply KDE to these $N$ predictions to find the 'center' $\mathbf{c}_{\text{KDE}}$ of the predictions by summing 2D Gaussian kernels (with a pre-defined variance $\Sigma$) centered at each prediction $\mathbf{c}^{(j)}$ to estimate a density $S(z; \mathcal{C})$:

$$\mathbf{c}_{\text{KDE}} = \underset{z \in [0,W] \times [0,H]}{\arg\max} S(z; \mathcal{C}) \quad \text{where} \quad S(z; \mathcal{C}) = \mathbb{E}_{\mathbf{c} \in \mathcal{C}} \left[ \exp\left(-\tfrac{1}{2}(z - \mathbf{c})^{\top} \Sigma^{-1} (z - \mathbf{c})\right) \right]. \quad (4)$$

A fixed-size bounding box $\mathcal{X}_{\text{RoI}}$ (with dimensions $W_{\text{RoI}} \times H_{\text{RoI}}$), centered at $\mathbf{c}_{\text{KDE}}$, defines the region of interest. We crop the image to this region: $\text{RoI}(x_{\text{img}}) = \text{crop}(x_{\text{img}}; \mathbf{c}_{\text{KDE}}, W_{\text{RoI}}, H_{\text{RoI}})$.

**Voting: Aggregating multiple votes within RoI.** In the voting stage, we further refine the coordinate prediction within this RoI. Given the local zoomed-in view, these predictions are expected to be more precise – the answer space is narrowed. We then reapply KDE to these new predictions within the RoI. This second application of KDE acts as a robust voting mechanism, aggregating the multiple refined predictions to determine the single coordinate with the highest probable point (i.e., the peak of the density). This two-stage searching strategy not only allows ReGUIDE to more precisely predict the final answer but also offers scalability: investing more computational resources in generating and evaluating samples for refinement generally leads to a more accurate result.

Formally, the model re-predicts $M$ new coordinates: $\mathcal{C}_{\text{RoI}} := \{\mathbf{c}_{\text{RoI}}^{(i)}\}_{i=1}^{N}$, where $(r^{(i)}, \mathbf{c}_{\text{RoI}}^{(i)}) \sim \mathcal{M}(\cdot \mid x_{\text{inst}}, \text{RoI}(x_{\text{img}}))$. Then, we use the same process as in Equation 4 to predict the most confident center point $\mathbf{c}_{\text{final}}$ within $\mathcal{C}_{\text{RoI}} \cup \mathcal{C}$ by robustly aggregating coordinates with KDE:

$$\mathbf{c}_{\text{final}} = \underset{z \in \mathcal{X}}{\arg\max} S(z; \mathcal{C}_{\text{RoI}} \cup \mathcal{C}). \quad (5)$$

## 4 Experiments

We provide an empirical evaluation of ReGUIDE by investigating the following questions:

- Can ReGUIDE enhance GUI grounding performance? (Table 1 and Table 3)
- Can ReGUIDE improve overall GUI agent performance? (Table 8)
- How does the MLLM perform reasoning based on the given instruction and image? (Table 4)
- Do the proposed components enhance the grounding performance? (Table 5 and Table 7)

Table 1: Accuracy (%) for ReGUIDE (Ours) and other baselines, for the fair comparison we trained with same dataset UGround with different data size. We evaluate on three web-grounding benchmarks: SCREENSPOT (ScrSpot), SCREENSPOT-V2 (ScrSpot-v2), and SCREENSPOT-PRO (ScrSpot-pro). Models trained with the same dataset but different size (i.e., a 20K subset of UGround, 10M fullset of UGround) . Bold indicates the best result within each group.

| Methods | Data Size | SCRSPOT | SCRSPOT-V2 | SCRSPOT-PRO |
|---|---|---|---|---|
| *Trained with same dataset* | | | | |
| UGround-2B (Gou et al., 2024) | 10M | 77.7 | 81.4 | 26.6 |
| UGround-7B (Gou et al., 2024) | 10M | 86.3 | 89.1 | 31.1 |
| Qwen-2.5-VL-3B (Alibaba, 2025) | - | 55.5 | 70.4 | 23.9 |
| + SFT | 20K | 56.8 | 56.5 | 11.6 |
| **+ ReGUIDE w/o TTS** | 20K | 84.9 | 87.6 | 27.9 |
| **+ ReGUIDE (Ours)** | 20K | **88.0** | **90.0** | **44.5** |
| Qwen-2.5-VL-7B (Alibaba, 2025) | - | 84.7 | 82.6 | 29.0 |
| + SFT | 20K | 84.9 | 88.9 | 25.9 |
| **+ ReGUIDE w/o TTS** | 20K | 88.1 | 91.0 | 36.3 |
| **+ ReGUIDE (Ours)** | 20K | **90.2** | **92.3** | **47.1** |

Before answering each question, we outline the experimental protocol (more details in Appendix A).

**Evaluation setup.** In the main results, we mainly report the grounding accuracy (%) as a metric. The prediction is counted as correct when the predicted point lies inside the ground-truth region. We evaluate ReGUIDE and baselines on SCREENSPOT (Cheng et al., 2024a), SCREENSPOT-V2 (Wu et al., 2024), SCREENSPOT-PRO (Li et al., 2025), evaluation benchmarks for GUI grounding. Especially, SCREENSPOT-PRO is a more challenging construct with a high-resolution image (i.e., up to $3840 \times 2160$). The agentic evaluation setting is in Section 4.3. During evaluation, we generate a prediction via greedy decoding. Specific test-time scaling setting is described in Appendix A.3

**Training setup.** For the main experiment, we train ReGUIDE on Qwen-2.5-VL 3B/7B (Alibaba, 2025). We utilize only a 20k subset of UGround (Gou et al., 2024) dataset, constituting approximately **0.2%** of its full set. First step of training for reasoning, we use the GRPO (DeepSeek, 2024) algorithm with two rewards, accuracy reward and $\lambda = 0.1$ weighted for formatting reward. For learning to predict consistent coordinates under transformations, where the local view is a random crop of up to 30% of the original area while preserving aspect ratio. Additionally, to demonstrate that ReGUIDE can be orthogonally applied on top of diverse base models, we perform consistency fine-tuning using the same training setup with only 2k dataset samples on several publicly available checkpoints, including UI-AGILE-7B (Lian et al., 2025), and Holo1.5-7B (H-Company, 2025). These results highlight that our consistency objective improves grounding performance regardless of the underlying model architecture or pretraining data. More setups in Appendix A.3.

**Baselines.** We compare our method against several baselines, which fall into two categories. First, we consider the model Qwen-VL-2.5 3B/7B supervised fine-tuned (SFT) on the identical dataset with ReGUIDE (i.e., 20k UGround subset). SFT optimises a coordinate-only loss and does not generate reasoning. Additionally, we consider proprietary models and GUI grounding models: GPT-4o (OpenAI, 2024), Claude3.7 (Anthropic., 2025), SeeClick (Cheng et al., 2024b), OS-Atlas-7B (Wu et al., 2024), AGUVIS-7B (Xu et al., 2024), UGround (Gou et al., 2024), and more.

## 4.1 MAIN RESULTS

As shown in Table 1, we present the main result by comparing the GUI grounding performance with other baselines. Here, we mainly compare ReGUIDE with open, closed models and SFT baseline.

**Comparison with controlled group.** First, compare models trained with the same base architecture (Qwen-2.5-VL) and dataset (a 20k subset of UGround). Our proposed method, ReGUIDE, significantly outperforms supervised fine-tuning (SFT) baselines across all evaluated scenarios. For instance, ReGUIDE achieves an accuracy of 88.0% on SCREENSPOT with the 3B model. Similar improvements are observed for the more challenging SCREENSPOT-PRO benchmark, where ReGUIDE boosts

Table 2: Accuracy (%) for other baselines and the model trained further with ReGUIDE. We evaluate on three web-grounding benchmarks: We evaluate on three web-grounding benchmarks: SCREENSPOT (ScrSpot), SCREENSPOT-V2 (ScrSpot-v2), and SCREENSPOT-PRO (ScrSpot-pro). The *Proprietary Models* include proprietary systems, while *Open-Sourced* covers public research models, The *GUI-trained + ReGUIDE* include the model trained further by ReGUIDE.

| Methods | Data Size | SCRSPOT | SCRSPOT-V2 | SCRSPOT-PRO |
|---|---|---|---|---|
| *Proprietary Models* | | | | |
| GPT-4o (OpenAI, 2024) | - | 18.3 | 16.6 | 0.8 |
| Claude 3.7 (Anthropic., 2025) | - | 82.1 | 87.6 | 27.7 |
| *Open-Sourced* | | | | |
| SeeClick (Cheng et al., 2024b) | 1M | 53.4 | 55.1 | 1.1 |
| OS-Atlas-7B (Wu et al., 2024) | 13M | 82.5 | 84.1 | 18.9 |
| AGUVIS-7B (Xu et al., 2024) | 1M | 84.4 | 85.8 | 22.9 |
| Aria-UI-7B (Yang et al., 2025b) | 1M | 82.4 | - | 11.3 |
| SE-GUI-7B (Yuan et al., 2025) | 3K | 88.2 | 90.3 | 47.3 |
| GUI-G2-7B (Tang et al., 2025) | 100K | 92.0 | 93.3 | 47.5 |
| GTA1-7B (Yang et al., 2025a) | 70K | - | 92.4 | 50.1 |
| UI-Venus-7B (Gu et al., 2025) | 107K | - | 94.1 | 50.8 |
| GUI-ARP-7B (Ye et al., 2025) | 5K | 89.3 | 91.8 | 60.8 |
| *GUI-trained + ReGUIDE* | | | | |
| UI-AGILE-7B (Lian et al., 2025) | 9K | 90.6 | 92.0 | 44.0 |
| **+ ReGUIDE (Ours)** | **+ 2K** | **92.0** | **92.8** | **50.8** |
| Holo1.5-7B (H-Company, 2025) | - | 91.7 | 93.9 | 57.8 |
| **+ ReGUIDE (Ours)** | **+ 2K** | **92.0** | **94.3** | **63.2** |

performance from 23.9% to 44.5%. Also in SCREENSPOT-V2 and dataset, ReGUIDE outperforms baseline and SFT model with a meaningful margin.

**Orthogonal adaptation on open-sourced models.** To further demonstrate the proposed consistency training and test-time scaling strategies can push existing models performance, we additionally apply ReGUIDE's consistency training fine-tuning with only additional 2k data samples and test-time scaling to several publicly availble base models, including UI-AGILE-7B, and Holo1.5-7B. As shown in Table 1, ReGUIDE consistently improves the grounding performance of these models. For example, UI-AGILE-7B improves from 44.0% to 50.8%, while Holo1.5-7B improves from 57.8% to 63.2% on SCREENSPOT-PRO. Also, the adapted Holo1.5-7B achieves the best performance among all 7B-sized models (63.2%). These results shows that the components of ReGUIDE are effective on various models and ReGUIDE can serve as a orthogonal mechanism that enhances existing GUI grounding systems.

**Comparison with open-sourced and proprietary models.** Beyond models trained with the same data and architecture, ReGUIDE achieves competitive or superior performance against to open-sourced and proprietary models. Specifically on SCREENSPOT, our 3B model surpasses UGround-7B (88.0 % > 86.3 %) and ReGUIDE's 7B model's show superior results (90.2 %) against other baselines. Also, ReGUIDE consistently outperforms other baselines in SCREENSPOT-V2, too. ReGUIDE shows the most remarkable performance on SCREENSPOT-PRO. ReGUIDE-7B performs accuracy of 47.1 %, significantly outperforming UGround-7B (31.1%). We report comparison against additional baselines in Appendix B.1, which also ReGUIDE consistently outperforms baselines. Importantly, when combine ReGUIDE 's training and KDE-inference scheme to other open-sourced 7B checkpoints such as Holo1.5-7B exhibit further gains (57.8% → 63.2%). The combined model achieves the highest accuracy among public 7B-scale GUI grounding models, demonstrating that ReGUIDE can elevate performance of existing systems despite variations in their data.

**Training Efficiency of ReGUIDE.** These results demonstrate that ReGUIDE not only surpasses supervised finetuned within the same training dataset but also establishes a new competitive standard against much larger, heavily trained open and closed models. Importantly, ReGUIDE achieves these improvements *using only a small fraction of the data (a 20k subset of UGround)* compared to other models (i.e., UGround: 10M, AGUVIS: 1M), highlighting its data efficiency and scalability. By

Table 3: Grounding Accuracy (%) for ReGUIDE (Ours) and UGround on the SCREENSPOT domain split. Results are broken down by device type, and UI element type. The right-most column reports the overall average. Bold indicates the best result within each column.

| Model | Data Size | Mobile | | Desktop | | Web | | Average |
|---|---|---|---|---|---|---|---|---|
| | | Text | Icon | Text | Icon | Text | Icon | |
| UGround-2B (Gou et al., 2024) | 10M | 89.4 | 72.0 | 88.7 | 65.7 | 81.3 | 68.9 | 77.7 |
| UGround-7B (Gou et al., 2024) | 10M | 93.0 | 79.9 | 93.8 | 76.4 | 90.9 | 84.0 | 86.3 |
| **ReGUIDE-3B** | 20K | 95.6 | 83.4 | 92.8 | 80.0 | 90.0 | 81.6 | 88.0 |
| **ReGUIDE-7B** | 20K | **95.6** | **85.6** | **94.3** | **85.7** | **92.6** | **84.5** | **90.2** |

Table 4: Example of ReGUIDE's generated reasoning and predicted coordinate.

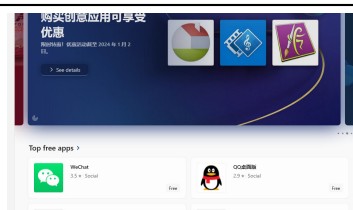

**Target Prompt:** Choose WeChat.

**Response:** `<think>` The task is to find the coordinate of the "WeChat" app icon in the image. The image shows a section labeled "Top free apps" with icons and names of apps. The "WeChat" app icon is located in the first row, first column of this section. `</think>` `<answer>(110,437)</answer>`

combining reinforcement learning, transform consistency learning, and scalable searching inference, ReGUIDE achieves a strong performance.

## 4.2 ADDITIONAL ANALYSIS AND ABLATION

**Generalizability over several domains.** We further analyze the generalizability of ReGUIDE across various GUI domains in multiple device environments (Mobile, Desktop, Web) and UI element types (Text, Icon). While our main results in Table 1 report the average accuracy on the SCREENSPOT, here we provide a more fine-grained evaluation. As shown in Table 3, ReGUIDE consistently achieves superior or competitive performance compared to the state-of-the-art open-sourced base model (i.e., Uground (Gou et al., 2024)). These results indicate that the training schemes of ReGUIDE effectively enhance its ability to generalize across diverse environments. Notably, ReGUIDE particularly shows strong performance for Icon elements within the Mobile and Desktop domains.

**Relative spatial reasoning from generation examples.** We analyze the reasoning path generated by ReGUIDE and the associated spatial predictions. First, we observe that the model naturally decomposes GUI localization tasks into interpretable reasoning steps, referencing relative spatial information. As shown in Table 4, the model explicitly identifies intermediate visual landmarks (e.g., a labeled section of "Top free apps") before specifying precise coordinates based on relative positions (e.g., "first row, first column"). This structured, relative reasoning pattern supports the models' ability to generalize across varying UI layouts.

**Ablation of training and inference components.** We perform an analysis on each component of ReGUIDE, specifically spatial reasoning (Reasoning), learn consistency under transformation (Consistency), and test-time searching (Searching). As shown in Table 5, each component plays an important role, leading to gradual and significant improvements when applied sequentially. Additional comparisons of alternative RL policy-optimization algorithms are provided in Appendix C.

**Effectiveness of Kernel Density Estimation.** To evaluate the effectiveness of Kernel Density Estimation (KDE) as an aggregating strategy, we conducted an ablation study using ReGUIDE-3B, comparing KDE against two alternatives: *Center* and *Medoid*. *Center* computes the mean of all predicted coordinates, while *Medoid* selects the prediction that minimizes the sum of distances to all others.

Table 6: Comparison of voting algorithm on SCREENSPOT and SCREENSPOT-PRO.

| Methods | SCREENSPOT | SCREENSPOT-PRO |
|---|---|---|
| Center | 80.3 | 20.2 |
| Medoid | 84.6 | 29.4 |
| KDE (Ours) | **88.0** | **44.5** |

As shown in Table 6, KDE significantly outperforms other voting strategies. This suggests that KDE provides a more stable aggregation by down-weighting the effect of outliers.

Table 5: Contribution of each proposed component of ReGUIDE on GUI grounding. We tested all three training components trained with Qwen-2.5-VL-3B: learn spatial reasoning (Reason), learn consistency under transformation (Consistency), and test-time spatial searching (Search). We report the GUI grounding accuracy (%) on SCREENSPOT and SCREENSPOT-PRO benchmarks.

| Model Size | Reason | Consistency | Search | SCREENSPOT | SCREENSPOT-PRO |
|---|---|---|---|---|---|
| | ✗ | ✗ | ✗ | 55.5 | 23.9 |
| | ✓ | ✗ | ✗ | 83.3 | 27.2 |
| 3B | ✓ | ✓ | ✗ | 84.9 | 27.9 |
| | ✓ | ✗ | ✓ | 85.2 | 40.7 |
| | ✓ | ✓ | ✓ | **88.0** | **44.5** |

Table 7: Contribution of each proposed component for test-time scaling, namely, the cropping and voting. We report the grounding accuracy (%) on SCREENSPOT and SCREENSPOT-PRO benchmarks with ReGUIDE-3B. The bold indicates the best results.

| Cropping | Voting | SCREENSPOT | SCREENSPOT-PRO |
|---|---|---|---|
| ✗ | ✗ | 84.3 | 27.9 |
| ✓ | ✗ | 81.7 | 42.7 |
| ✓ | ✓ | **88.0** | **44.5** |

**Ablation of inference components.** Moreover, we perform ablation to prove the effectiveness of each component in the two-stage inference time searching strategy. As shown in Table 7, both the crop stage and the voting stage contribute to improvement in performance. It is notable that in SCREENSPOT-PRO, the improvement induced by crop stage is huge, which indicates that localization plays a crucial role for the proper understanding of high-resolution images.

**Internal attention after consistency finetuning.** To assess how the consistency-under-transformation stage alters internal representations, we visualised the average attention logits of the last transformer layer in ReGUIDE-3B. As shown in Figure 2, the GRPO-only concentrates most of its attention on the reasoning span, which indicates limited contextual grounding. In contrast, the ReGUIDE model, after consistency finetuning, distributes attention more evenly across reasoning and other text tokens. This result imply that consistency finetuning enables richer spatial awareness and reduced attention collapse.

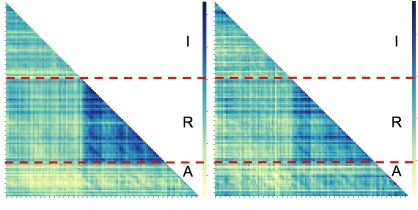

(a) GRPO trained    (b) ReGUIDE

Figure 2: Attention map of text tokens. *I*: instruction, *R*: rationale, and *A*: answer.

**Natural Gaussian distribution of coordinate tokens likelihood.** As an additional qualitative analysis, we examine the likelihood distribution of coordinates in the neighborhood of the initial predicted point, conditioned on the same generated reasoning. As shown in Figure 3, theses likelihoods form a surprisingly smooth, Gaussian-shaped distribution. This Gaussian pattern suggests that the ReGUIDE implicitly captures continuous spatial uncertainty, despite operating in discrete language token space. The observation provides a strong empirical motivation for our proposed Gaussian-weighted inference strategy (i.e., KDE), which leverages them to improve coordinate prediction accuracy and robustness. Seemingly, a concurrent work GUI-G2 (Tang et al., 2025) also suggested a Gaussian-shaped structure, but in a fundamentally difference purpose: GUI-G2 introduces a Gaussian reward during training, inspired from findings in human cursor control research (Fitts, 1954). However, ReGUIDE leverages the Gaussian structure explicitly at test-time through our scaling procedure rather than using it for training. Notably, our empirical observation not only supports the design choice of our KDE-based inference, but also suggests a potential explanation for why Gaussian-based reward shaping, as used in GUI-G2, may be inherently effective.

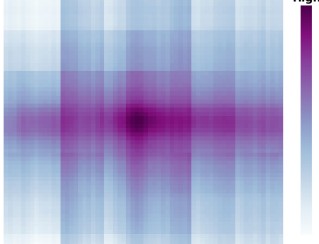

Figure 3: Likelihood of coordinate language tokens.

**Ablation on inference time searching strategy.** We perform ablation for three hyperparameters on inference time scaling strategy, that is, crop size ($W_{\mathtt{RoI}}$), generation samples ($N$), and temperature ($T$). As shown in Figure 4, increasing $N$ gradually improves the grounding performance, demonstrating that our strategy is scalable. We adopt the default hyperparameters $N = 16$, $L = 840$, and $T = 1.0$.

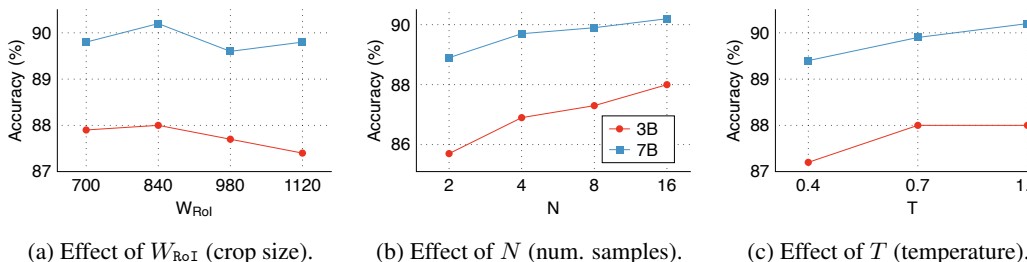

(a) Effect of $W_{\texttt{RoI}}$ (crop size).      (b) Effect of $N$ (num. samples).      (c) Effect of $T$ (temperature).

Figure 4: Ablation studies on crop size $W_{\texttt{RoI}}$, number of samples $N$, and decoding temperature $T$.

Table 8: Task-success rate (%) on the two offline-agent task benchmarks, i.e., Multimodal-Mind2Web (MM2W), and Android Control (AC). For Multimodal-Mind2Web, there are 3 test sets cross-domain, cross-task, and cross-web. For Android Control, there are 2 test sets low and high.

| Grounding Module | Data Size | MM2W | | | AC | |
|---|---|---|---|---|---|---|
| | | domain | task | web | low | high |
| UGround-2B (Gou et al., 2024) | 10M | 47.7 | 48.6 | 47.6 | 65.0 | **50.0** |
| UGround-7B (Gou et al., 2024) | 10M | 48.5 | 50.7 | 48.1 | 66.2 | 49.8 |
| **ReGUIDE-3B (Ours)** | 20K | 48.8 | 50.5 | 47.7 | 66.2 | 49.8 |
| **ReGUIDE-7B (Ours)** | 20K | **49.5** | **52.0** | **48.7** | **67.4** | **50.0** |

## 4.3 AGENTIC SETTINGS EXPERIMENTS

To verify whether better groun ding transfers to real tasks, we plug each grounding model with ReGUIDE and use a high-level planner (GPT-4o). For every episode, the planner emits a natural-language action plan; the grounding module converts each step into pixel coordinates. Here, we report task-success rate (%) on five offline suites: Multimodal-Mind2Web (Deng et al., 2023), AndroidControl (Li et al., 2024). All prompts, settings, and frameworks are followed by UGround (Gou et al., 2024) and the detailed settings are described in Appendix A.

As shown in Table 8, improved grounding performance made by ReGUIDE leads to better agent performance. With only 3B parameters, ReGUIDE matches or edges out the larger UGround-7B on Multimodal-Mind2Web and Android Control-High. The 7B variant of ReGUIDE consistently outperforms other baselines (i.e., +1.3% in Multimodal-Mind2Web cross task). These gains, achieved with identical planners and action budgets, substantiate that the grounding improvements provided by self-evolutionary RL and global–local consistency directly enhance full-task completion.

## 5 DISCUSSION AND CONCLUSION

We proposed ReGUIDE, a novel and effective GUI grounding framework that significantly enhances the capabilities of Multimodal Large Language Models by enabling data-efficient learning through self-generated reasoning and spatial-aware criticism. Our key idea is leveraging online reinforcement learning for self-generating language reasoning and employing spatial priors to criticize predictions, and further boosting performance at inference time through a test-time scaling search strategy that integrates spatial search with coordinate aggregation. We demonstrated that ReGUIDE consistently outperforms other open-sourced baselines, even when trained with a tiny fraction of data, such as only 0.2% of the samples used by the best open-sourced baselines. Crucially, these advances in data-efficient grounding translate to improved performance in downstream agentic tasks, highlighting ReGUIDE's potential to develop more capable and practical GUI agents.

**Future works and limitations.** We believe it will be an interesting future direction to train MLLM planners that can operate hierarchically with ReGUIDE, fully capitalizing on its precise grounding performance. Additionally, a potential limitation is that while ReGUIDE is a very effective method, it lacks an explicit safety framework to prevent malicious uses, such as hacking or spreading misinformation—a challenge common to many grounding models. However, future investigations could explore the research and development of robust safeguards.

ETHICS STATEMENT

This paper presents ReGUIDE, a method that significantly improves GUI-grounding accuracy. We expect that our approach will enhance digital accessibility and productivity by enabling agents to locate and interact with on-screen elements more reliably. However, the same capability could be misused for malicious purposes, such as automated hacking or spreading misinformation, issues that current agent models already face. Mitigation measures include responsible release practices, access controls, and detection tools to discourage adversarial deployments.

REPRODUCIBILITY STATEMENT

We have made extensive efforts to ensure the reproducibility of our results. Conceptual and experimental details are provided in Section 4, including dataset descriptions and evaluation setups. Training configurations, optimization hyperparameters, and evaluation protocols are described in Appendix A.3, while details of the computational resources used are summarized in Appendix A.4. To further facilitate reproducibility, we provide our implementation and scripts in the supplementary materials, which include the codebase for training, evaluation, and inference. Together, these resources should allow researchers to reproduce our experiments and follow the reported results.

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

# A EXPERIMENTAL DETAILS

In this section, we describe the experimental details of Section 4, including ReGUIDE and baselines.

## A.1 DATASET DETAILS

In this section, we describe the dataset we used in training and evaluation.

- **UGround** Uground is a Graphical User Interface (GUI) grounding dataset that consists of 10M triplets of GUI image, element coordinates, and description of the element, which was synthesized by open MLLM, Llava-Next-13 B. Uground (Gou et al., 2024) models are trained with their own synthesized web GUI dataset, which consists of 10 M image-instruction pairs. For training ReGUIDE, we randomly select a 20K image-instruction set from UGround (Gou et al., 2024) dataset, which is approximately 0.2% of the original dataset.

- **SCREENSPOT** ScreenSpot (Cheng et al., 2024a) is an evaluation benchmark dataset which consists of 1,272 (image, description, bounding box) triplets. There are various GUI domains in multiple device environments (Mobile, Desktop, Web) and UI element types (Text, Icon).

- **SCREENSPOT-V2** Wu et al. (2024) refined the original ScreenSpot dataset by resolving ambiguous descriptions and aligning annotations more precisely with visual elements. We use both versions for a more accurate grounding evaluation.

- **SCREENSPOT-PRO** ScreenSpot-Pro (Li et al., 2025) is a benchmark designed to evaluate GUI grounding models in professional, high-resolution environments. It spans 23 applications across five professional categories and three operating systems, highlighting the challenges models face when interacting with complex software..

- **Android Control** Android Control (Li et al., 2024) is an offline agent benchmark comprising 15000 distinct user tasks sampled from 833 different Android applications. Each task contains a sequence of screenshots, detailed action records, and accessibility trees derived from human demonstrations (the "golden trajectory"). Each action is also annotated with a specific instruction (e.g., "set the hours to 6"), allowing both high-level and low-level task settings. In our experiments, we followed Gou et al. (2024) and evaluated models on 500 randomly sampled steps from the test split.

- **Multimodal-Mind2Web** The Multimodal-Mind2Web (Deng et al., 2023) dataset contains screenshots with a large vertical dimension (e.g., 1280 x 10000 pixels). The test split includes 1,013 realistic user tasks collected from over 100 different websites, each accompanied by a high-level instruction and a sequence of action steps. To make the data manageable we followed (Gou et al., 2024), and divide the image vertically with certain amount of duplication. During the agent evaluation, when an agent cannot find an actionable element or explicitly chooses to scroll, the next block is processed, simulating user scrolling.

## A.2 MODEL DETAILS

- **Qwen-2.5-VL** We use Qwen-2.5-VL (Alibaba, 2025) is a multimodal large language model (mllm) which inputs text and images and outputs text. The model is equipped with vision-language fusion modules and trained on diverse visual instruction datasets. We trained ReGUIDE from Qwen-2.5-VL-3B and 7B models.

- **UGround** UGround (Gou et al., 2024) is the model trained from Qwen-2-VL (Alibaba, 2025), there are two versions UGround-2B and UGround-7B. They are trained with a synthesized dataset that has 10M images, instructions, and target coordinates triplets. This amount of dataset size is about 500 times bigger than the ReGUIDE trained. For the fair comparison, we trained UGround on Qwen-2.5-VL (Alibaba, 2025) with the same subset used to train ReGUIDE. As shown in Table 1, ReGUIDE excessively outperforms UGround for both.

- **Proprietary Models Endpoint**
  - *gpt-4o-2024-05-13*
  - *claude-3-7-sonnet-20250219*

## A.3 Training and Evaluation Details

- **Training framework** For the training framework, we utilize VERL for reinforcement learning and TRL for fine-tuning.
  - TRL (https://github.com/huggingface/trl)
  - VERL (https://github.com/volcengine/verl)
- **Training hyperparameters.** We describe our training hyperparameters, which we used for training, especially for learning to explaining GUI images via reasoning and learning to predict consistent coordinates under transformations.

Table 9: Hyperparameters for ReGUIDE on grpo training

| Hyperparmeter | Value |
|---|---|
| Optimizer | Adam |
| Algorithm | GRPO |
| Learning rate | 1e-6 |
| Training data size | 20k |
| Batch size | 128 |
| Generation per Sample | 8 |
| kl_coef | 0.01 |

Table 10: Hyperparameters for ReGUIDE on training consistent coordinated under transformations

| Hyperparmeter | Value |
|---|---|
| Optimizer | Adam |
| Learning rate | 1e-6 |
| Training data size | 20k |
| Epoch | 1 |
| Batch size | 64 |
| minimum crop ratio | 0.3 |

## A.4 Compute Resource

For the main development, we mainly use Intel(R) Xeon(R) Gold 6338 CPU @ 2.00GHz and four A100 80GB GPUs. For training ReGUIDE, it took around 20 hours.

## A.5 Test-time Scaling Strategy Setting

For the test-time scaling strategy, if there is further mention about hyperparmeter, we used prediction sampling $N = M = 16$ samples with temperature $T = 1.0$, crop bounding box size as $W_{\texttt{RoI}} = H_{\texttt{RoI}} = 840$, and KDE variance $\Sigma = 0.01$.

# B ADDITIONAL EXPERIMENTAL RESULTS

## B.1 COMPARISON WITH ADDITIONAL BASELINES

Table 11: Accuracy (%) for ReGUIDE (Ours) and other baselines, including UI-TARS, UI-R1, GUI-R1, and ReGUIDE (Ours). We evaluate on three web-grounding benchmarks: SCREENSPOT, SCREENSPOT-V2, and SCREENSPOT-PRO.

| Methods | SCREENSPOT | SCREENSPOT-V2 | SCREENSPOT-PRO | Average |
|---|---|---|---|---|
| UI-R1-E-3B | 89.2 | 89.5 | 33.5 | 70.7 |
| GUI-R1-7B | - | - | 31.3 | - |
| UI-TARS-2B | 82.3 | 84.7 | 27.7 | 64.9 |
| UI-TARS-7B | 89.5 | 91.6 | 35.7 | 72.3 |
| Qwen2.5-VL-7B + *Region Focus* | - | - | 32.1 | - |
| UI-TARS-7B + *Region Focus* | - | - | 41.2 | - |
| **ReGUIDE-3B (Ours)** | 88.0 | 90.0 | 44.5 | 74.2 |
| **ReGUIDE-7B (Ours)** | 90.2 | 92.3 | 47.1 | 76.5 |

We compare the grounding accuracy of ReGUIDE with other additional baselines, which are closed-source (UI-TARS (Qin et al., 2025)) and the concurrently proposed reinforcement learning methods for grounding UI-R1 (Lu et al., 2025) and GUI-R1 (Xia & Luo, 2025). As shown in Table 11 ReGUIDE achieves the highest average combined accuracy (75.3%) across SCREENSPOT (Cheng et al., 2024a), SCREENSPOT-V2 (Wu et al., 2024), and SCREENSPOT-PRO (Li et al., 2025).

Crucially, the compact ReGUIDE-3B already surpasses the best 7B-parameter base model (UI-TARS-7B) by +1.8 pp in average accuracy (72.3 % → 74.1 %), demonstrating that our training and search strategy substantially improves the GUI grounding performance. Moreover, both reinforcement fine-tuned baselines (i.e., UI-R1-E-3B, GUI-R1-7B) perform worse than ReGUIDE, underscoring the effectiveness of our further training for consistency under transformation and search algorithm. The full-sized ReGUIDE-7B further extends this lead to +4.2 pp, establishing a best among the given baselines on the combined benchmarks.

Moreover, we also compare ReGUIDE with the recent visual test-time scaling method, Region Focus (Luo et al., 2025), which improves grounding by repeatedly focusing on local regions and re-predicting actions. In contrast, ReGUIDE adopts a different approach based on cropping and aggregating predictions. As analyzed in Table 6, we systematically evaluated several statistical aggregation strategies and found that Kernel Density Estimation (KDE) yields the most reliable results. This choice is further motivated by the Gaussian-shaped likelihood distribution observed in Figure 3, which provides a natural justification for KDE as an aggregation method. Despite using a smaller backbone, ReGUIDE-3B consistently outperforms Region Focus even against their 7B models, highlighting both the effectiveness and efficiency of our approach.

## B.2 EFFECT OF CONSISTENCY FINETUNING AFTER REINFORCEMENT STAGE

Table 12: Effect of consistency-under-transformation finetuning on ReGUIDE-3B, evaluated on the SCREENSPOT (Cheng et al., 2024a) benchmark.

| Method | Augmentation usage | Accuracy (%) |
|---|---|---|
| RL w/o Aug. | – | 83.3 |
| RL w/ Aug. | In RL stage | 83.2 |
| ReGUIDE (Ours). | fine-tune | **84.9** |

We assess the benefit of our *consistency-under-transformation* finetuning on the ReGUIDE-3B model using the ScreenSpot benchmark. As shown in Table 12, training GRPO on the original screenshots only (RL w/o Aug.) achieves 83.3 % accuracy, while mixing the same augmented views directly into the RL phase (RL w/ Aug) yields no improvement (83.2 %). In contrast, our two-stage pipeline—first training GRPO without augmentation and then applying a separate consistency finetune on the augmented views (ReGUIDE)—raises accuracy to 84.9 %, a gain of +1.6 pp over the vanilla RL

model and +1.7 pp over RL w/ Aug. This result confirms that the performance boost arises from explicitly learning to align predictions across transformed views, not from data augmentation alone.

## B.3 DATASET SCALABLE EXPERIMENT

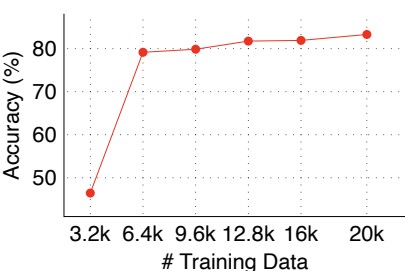

Figure 5: Grounding accuracy (%) on SCREENSPOT (Cheng et al., 2024a) as the UGround (Gou et al., 2024) training set grows from 3.2 K to 20 K samples. We trained from Qwen-2.5-VL-3B.

To measure how our reinforcement-learning stage benefits from more data, we varied the amount of UGround (Gou et al., 2024) training samples from 3.2K up to 20K in 3.2K increments and evaluated grounding accuracy (%) on the SCREENSPOT (Cheng et al., 2024a) benchmark (Figure 5). Accuracy rises monotonically with data size, and the full 20K subset yields the best performance. These results indicate that the ReGUIDE training pipeline scales effectively with additional data and has not yet saturated at the 20K level.

## B.4 ONLINE AGENT EVALUATION

Table 13: Task success rate (%) on Android World.

| Methods | Android World |
|---|---|
| UGround-7B | 44.0 |
| ReGUIDE-3B | 46.1 |

We further evaluate ReGUIDE on the Android World (Rawles et al., 2024) online agent benchmark, comparing the grounding accuracy (%) of ReGUIDE-3B with UGround-7B (Gou et al., 2024). As shown in Table 13, ReGUIDE-3B surpasses UGround-7B by +2.1%, despite having considerably fewer parameters. This finding indicates that stronger grounding performance directly translates into improved results on complex online agent evaluations.

## B.5 KDE-BASED TEST-TIME SCALING.

Table 14: Accuracy (%) for baselines including UGround-7B and GTA1-7B with KDE test-time scaling. We evaluate on two web-grounding benchmarks: SCREENSPOT-V2, and SCREENSPOT-PRO.

| Methods | SCREENSPOT-V2 | SCREENSPOT-PRO |
|---|---|---|
| UGround-7B | 89.6 | 40.3 |
| + KDE-TTS | 91.0 | 46.6 |

We additionally evaluate the proposed KDE-based test-time scaling on several opensource base model to measure its standalone effectiveness. As shown in the Table 14, KDE-base scaling consistently improves the grounding accuracy of the model (i.e., UGround-7B) across both ScreenSpot-v2 and ScreenSpot-pro. For example, UGround -7B improves from 40.4% to 46.6% on SCREENSPOT-PRO. These results confirm that KDE-based search works as a model-agnostic inference enhancement.

Table 15: Comparison of the average consuming prediction time and grounding accuracy (%) on SCREENSPOT and SCREENSPOT-PRO benchmarks with ReGUIDE-3B. N denotes the number of sampling during test-time.

| Method | Scaling | N | SCREENSPOT | | | SCREENSPOT-PRO | | |
|---|---|---|---|---|---|---|---|---|
| | | | Time (s) | Time Cost | Accuracy | Time (s) | Time Cost | Accuracy |
| ReGUIDE-3B | ✗ | - | 1.69 | x1.0 | 84.9 | 2.43 | x1.0 | 27.9 |
| | ✓ | 4 | 3.06 | x1.8 | 86.9 | 3.98 | x1.6 | 41.1 |
| | ✓ | 8 | 3.31 | x2.0 | 87.3 | 4.29 | x1.8 | 43.5 |
| | ✓ | 16 | 3.76 | x2.2 | 88.0 | 4.88 | x2.0 | 44.5 |

### B.6 COMPUTATION COST OF THE INFERENCE

We measured the average elapsed time for inference of samples in ReGUIDE and compared between with and without test-time inference procedure. We used a single A100 80G GPU for fair comparison, and employed the vLLM (Kwon et al., 2023) framework for inference. To ensure fairness, we warmed up the pipeline with 25 preliminary samples before measuring the average inference time over the next 25 samples, following the recommendation in (Zhao et al., 2024).

As shown in the Table 15, the full test-time procedure (N=16) takes roughly twice as long as single-pass baselines, which is expected since the crop-and-vote pipeline requires two inference passes. Nevertheless, by leveraging vLLM's continuous batching capability and caching identical conditioned images and questions, our system maintains high efficiency.

In the aspect of user experience, an important strength of ReGUIDE is that its test-time scaling offers a adjustable trade-off between latency and accuracy. While the full-performance yields the highest performance at approximately 2x latency, users can choose for lighter configuration. For example, reducint the number of samples from N=16 to N=4 decreases latency by about 64% while retaining roughly 80% of the performance improvement. This flexibility suggests end-users and practitioners to control ReGUIDE's efficiency-effectiveness trade-off based deployment. Moreover, while several recent approaches (Lian et al., 2025; Li et al., 2025) that rely on external APIs, which generate network/server-bound latency, ReGUIDE operates entirely locally, avoiding additional overhead.

## C ALTERNATIVE RL POLICY OPTIMIZATION ALGORITHMS

Table 16: Comparison of RL algorithms. We report accuracy (%) and average output length (Output Len.) on ScreenSpot Benchmark (Cheng et al., 2024a).

| Methods | Accuracy (%) | Output Len. |
|---|---|---|
| REINFORCE++ (Hu, 2025) | 64.8 | 32.3 |
| RLOO (Ahmadian et al., 2024) | 77.8 | 95.7 |
| GRPO (DeepSeek, 2024) | 83.3 | 80.8 |

While we primarily adopt GRPO (DeepSeek, 2024) for the learning to visual reasoning, alternative RL algorithms, such as RLOO (Ahmadian et al., 2024), REINFORCE++ (Hu, 2025), could also be considered. To verify the performance difference between RL poly optimization algorithms, we compared grounding accuracy (%) on SCREENSPOT. As shown in the Table 16, GRPO achieves the highest accuracy (83.3%), followed by RLOO (77.8%), while REINFORCE++ significantly underperforms (64.8%). Interestingly, we observed that the REINFORCE++ trained model loses its reasoning ability (i.e., shortest output length), eventually failing to generate meaningful reasoning paths.

## D ILLUSTRATION OF SEARCHING STRATEGY.

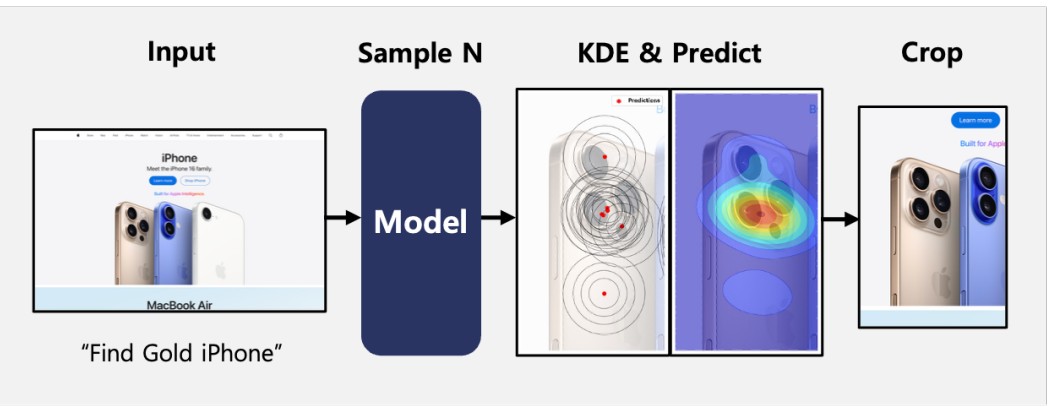

Figure 6: **Overview of ReGUIDE's searching strategy.**. Starting from the full screenshot, the model samples $N$ coordinate predictions, votes using kernel density estimation, and recenters a crop on the density peak. One searching pass is then run inside the crop, and the point with the highest KDE density is returned as the final prediction.

# E LEARNING CURVE

We report the learning curve of ReGUIDE during training GRPO. The base model of the training curve is Qwen-2.5-VL-3B. We can see that the reward is kept increasing during the training. Also, the mean response length has decreased during the training.

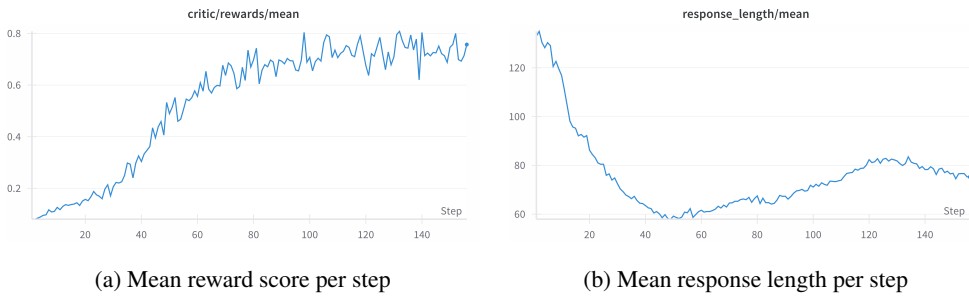

(a) Mean reward score per step   (b) Mean response length per step

Figure 7: Learning curve during training GRPO of ReGUIDE.

# F PROMPTS

We describe our generation prompt for ReGUIDE (in Table 17 and UGround (Gou et al., 2024) (in Table 18. We used the ReGUIDE prompt for training and evaluation ReGUIDE, and the UGround prompt for supervised fine-tuning the Qwen-2.5-VL for comparison and evaluating the UGround model.

Table 17: Prompt of ReGUIDE

---

*Element*: Target element instruction.
*size_x*: Width of the given image.
*size_y*: Height of the given image.

---

What is the coordinate of [{*Element*}] in the image?
The size of image is (*{size_x},{size_y}*).
Output the thinking process in <think> </think> and final answer (coordinate (x,y)) in <answer> </answer> tags.

---

Table 18: Prompt of UGround

---

*Element*: Target element instruction.

---

Your task is to help the user identify the precise coordinates (x, y) of a specific area/element/object on the screen based on a description.

- Your response should aim to point to the center or a representative point within the described area/element/object as accurately as possible.
- If the description is unclear or ambiguous, infer the most relevant area or element based on its likely context or purpose.
- Your answer should be a single string (x, y) corresponding to the point of the interest.

Description: {*Element*}

Answer:

---

## G  GENERATION EXAMPLES

Table 19: Examples of ReGUIDE's generated reasoning and predicted coordinate.

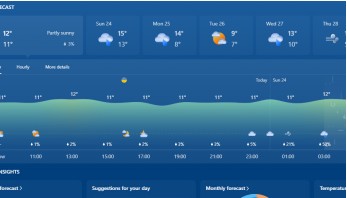

**Target Prompt:** add description.

**Response:** <think>To find the coordinate of [add description], I need to locate the element labeled Ädd descriptionïn the image. The coordinates are given as (x, y) pairs, and I need to identify the position of this element.</think><answer>(300,2221) </answer>

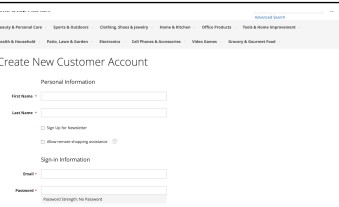

**Target Prompt:** check weather on tuesday.

**Response:** <think> To determine the coordinate of the ïlpfolder, I need to locate it in the image and identify its position relative to the other folders. The ïlpfolder is the second folder in the top row, as indicated by its position in the grid layout. </think> <answer>(584,760)</answer>

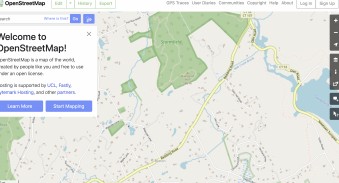

**Target Prompt:** sign up for newsletter.

**Response:** <think> The task is to find the coordinate of the Šign Up for Newslettercheckbox in the image. The checkbox is located in the Personal Informationsection, and it is a small square with a label next to it. The coordinates are given in a specific size of the image, and we need to identify the position of the checkbox within the image. </think> <answer>(367,85)</answer>

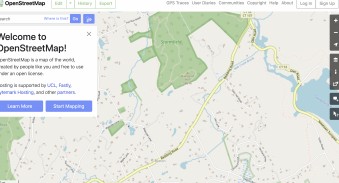

**Target Prompt:** start mapping.

**Response:** <think> The Štart Mappingbutton is typically used to begin the process of adding or editing data on the map. In the image, it is located in the lower left section, which is a common placement for such buttons on web interfaces. </think> <answer>(558,805)</answer>

Table 20: Failure types and their frequency from sampled generation.

| Failure Type | Number |
|---|---|
| Understand | 12 |
| Reasoning | 12 |
| Localization | 6 |
| Total | 30 |

## H  FAILURE ANALYSIS

We conducted a failure analysis of sampled 30 error cases generated by ReGUIDE-3B on the ScreenSpot. Then, we categorized failure cases into 3 groups, (i) failure to understand or perceive the target, (ii) incorrect spatial reasoning, and (iii) incorrect localization. As shown in the Table 20, interestingly, 20% (=6/30) of cases are corrected in reasoning but fail to localize correctly. This highlights an important area for future work to improve localization results, followed by correct reasoning.

