# OpenReview forum: "ReGUIDE: Data Efficient GUI Grounding via Spatial Reasoning and Search"
_ICLR.cc/2026/Conference — Submitted to ICLR 2026_

### Official Review · Reviewer_6X4C · 2025-10-21

**Soundness:** 3
**Presentation:** 3
**Contribution:** 2
**Rating:** 4
**Confidence:** 5

**Summary:**

This paper presents ReGUIDE, a data-efficient GUI grounding framework for MLLM-based web agents. It combines reasoning-guided self-generation and spatial-aware criticism to improve coordinate localization without large datasets. A test-time spatial scaling further enhances accuracy. Compared with exisiting grounding models, ReGUIDE perfroms well on extensive expermients results.

**Strengths:**

1. The paper provides an extensive and comprehensive evaluation of ReGUIDE across multiple benchmarks.
2. The writing and organization are clear, coherent, and well-structured.
3. The proposed training and testing time scaling strategies are effective, and the combination of global–local search with voting demonstrates strong performance.

**Weaknesses:**

1. The main concern with this work lies in whether the proposed model truly achieves state-of-the-art grounding performance. In fact, several prior studies have already reported superior results. However, the authors did not include these stronger baselines (e.g., [1–3]) in their benchmark comparisons.

2. The ensuing concern is that the related work does not encompass the latest developments.

3. The results presented in Table 7 seem somewhat counterintuitive. First, it is unclear why five data benchmarks are listed, as in reality there are only two. Second, regarding AC benchmark, it is puzzling that the High-level model outperforms the Low model. Finally, it remains unclear how the experimental setup allows web agents to achieve such high performance on mobile tasks. If additional fine-tuning was applied, the reported task completion rate still appears lower than that of some existing mobile agents.

4. The relationship between the effect of consistency fine-tuning and attention–grounding modeling remains unclear. The authors should further clarify how these changes contribute to improved grounding performance.

5. The analysis of the natural Gaussian distribution of coordinate token likelihood appears to yield conclusions similar to those reported in GUI-G2. Therefore, it is strongly recommended that the authors compare their findings directly with GUI-G2, clarifying any differences in methodology, experimental setup, and quantitative outcomes.

**Reference**

[1] GTA1: GUI Test-time Scaling Agent

[2] GUI-G2: GAUSSIAN REWARD MODELING FOR GUI GROUNDING

[3] Enhancing Visual Grounding for GUI Agents via Self-Evolutionary Reinforcement Learning

**Questions:**

See above

---

> ### Author Response · Authors · 2025-11-22
> **Response to Reviewer 6X4C (1/2)**
>
> Dear reviewer 6X4C,
>
> We sincerely appreciate your valuable comments. We respond to your questions below.
>
> ---
>
> **[W1, W2] Comparison with other baselines (GTA1-7B, GUI-G2, and SE-GUI-7B).**
>
> We carefully remark that we already discussed and compared several baselines (e.g., UGround, AGUVIS, SeeClick,  UI-R1, GUI-R1) with ReGUIDE. Nonetheless, thanks to the reviewer, we agree that including discussion and comparison about such concurrent works (GTA1-7B, GUI-G2, and SE-GUI-7B) will make the paper clearer.
> Conceptually, these concurrent approaches (GTA1-7B, GUI-G2, and SE-GUI-7B) mainly focused on designing the reward and composing a dataset effectively. In contrast, ReGUIDE is trained on 20k randomly sampled dataset and uniquely introduces (i) consistency learning aligns global-local spatial prior and (ii) KDE-based spatial search which is orthogonal to the standard RL approaches used in GTA1-7B, GUI-G2, and SE-GUI-7B.
>
> Regarding direct numerical comparison, since concurrent works are trained with heterogeneous dataset, we believe that raw numbers should be interpreted with care. To address this issue more constructively, we focus on verifying whether ReGUIDE’s unique components are complementary to concurrent baselines. To this end, we continually fine-tune GTA1-7B which is the strongest among the baselines cited by the reviewer. By applying ReGUIDE’s components to the official GTA-1 checkpoint, we achieved a performance gain of 2.7%p (50.7 -> 53.4) on ScreenSpot-pro. This result confirms that ReGUIDE effectively complements the other baselines.
>
> Furthermore, to demonstrate SOTA of ReGUIDE, we additionally adapted ReGUIDE to Holo1.5-7B, which is currently on the strongest open-source baselines. As shown in the table below, **adapted Holo1.5-7B achieved 63.2 on ScreenSpot-pro, a state-of-the-art performance** among open-sourced 7B-scale models.
>
> These results demonstrate that ReGUIDE is not outdated, and can reliably enhance existing GUI grounding methods and push the grounding performance to SOTA.
>
>
> \begin{array}{lcccc}\hline
> \text{Model} & \text{ScreenSpot} & \text{ScreenSpot-v2} & \text{ScreenSpot-Pro} \newline \hline
> \text{GTA1-7B} & - & 92.0 & 50.7  \newline
> \text{  + ReGUIDE} & - & \textbf{92.8} & \textbf{53.4} \newline \hline
> \text{UI-AGILE-7B} & 90.6 & 92.0 & 44.0 \newline
> \text{  + ReGUIDE} & \textbf{92.0} & \textbf{92.8} & \textbf{50.8} \newline \hline
> \text{Holo1.5-7B}  & 91.7 & 93.9 & 57.8 \newline
> \text{ + ReGUIDE}  & \textbf{92.0} & \textbf{94.3} & \textbf{63.2} \newline \hline
> \end{array}
>
> We thank the reviewer for a valuable question and have incorporated these results in the revised manuscript (highlighted in blue).
>
> [1] Lian et al., Ui-agile: Advancing gui agents with effective reinforcement learning and precise inference-time grounding, preprinted arxiv2507\
> [2] Hcompany, Holo1.5 - Open Foundation Models for Computer Use Agents, huggingface\
> [3] Yang et al., GTA1: GUI Test-time Scaling Agent, preprinted arxiv2507
>
>
>
> ---
>
> **[W3] More explanation about Table 7 contents.**
>
>
> Thank you for pointing this out. There was little mistake about column names. We switch the column name high and low. Also, as you said there are two benchmarks with different sub tasks (cross-domain, cross-task, cross-web, android control-high, android control-low). To clarify the reading of the model we revise the caption and explanation of the benchmark.
>
> In the aspect of high performance on mobile, we didn’t additionally train for the task of the Android Control. We trained the dataset which is a subset of UGround [4] which contains a small amount of mobile screen-size data which could impact the performance in such mobile tasks.
>
> [4] Gou et al., Navigating the Digital World as Humans Do: Universal Visual Grounding for GUI Agents, ICLR2025
>
> ---

---

> ### Author Response · Authors · 2025-11-22
> **Response to Reviewer 6X4C (2/2)**
>
> **[W4] Relationship between consistency fine-tuning and test-time scaling method.**
>
> We thank the reviewer for the insightful question. We interpret "attention–grounding modeling" as our KDE-based Test-Time Scaling (TTS) strategy, which means focusing on local regions (cropping).
>
> We clarify that we have already investigated the relationship between the effect of consistency fine-tuning and attention–grounding modeling in our ablation study (Table 5 of the original manuscript). To explicitly show how Consistency Fine-tuning contributes to Attention-Grounding (TTS) performance, we compare the results with and without Consistency training under the same TTS setting. As shown in the summarized table below, the model trained **without consistency achieves 40.7% (+13.5%p)** on ScreenSpot-Pro, whereas the **consistency-trained model reaches 44.5%(+17.3%p)**. This means that consistency learning increases the effectiveness of test-time scaling by **approximately 28% (17.3 vs 13.5)**. This substantial difference indicates that the internal representation learned through consistency learning makes the test-time spatial search significantly more effective.
>
> We believe this behavior is consistent with prior consistency-based robustness methods [4,5,6], which report that enforcing across augmented views stabilizes the representation under input perturbations. In our setting, the global screen and its local crops can be viewed as spatial augmentations. We conjecture that Consistency Learning encourages scale-robust spatial features, allowing the cropping-based search to exploit them more effectively.
>
> \begin{array}{cccc}\hline
> \text{RL} & \text{consistency} & \text{TTS} & \text{ScreenSpot-pro} \\ \newline\hline
> O&X& X & 27.2 \\ \newline
> O&X& O & 40.7 \\ \newline
> O&O& O & 44.5 \\ \newline
> \hline
> \end{array}
>
>
> [5] Hendricks et al., AugMix: A Simple Data Processing Method to Improve Robustness and Uncertainty, ICLR 2020 \
> [6] Zhang et al., MEMO: Test Time Robustness via Adaptation and Augmentation, Neurips 2022 \
> [7] Xie et al., Unsupervised Data Augmentation for Consistency Training, Neurips 2020
>
>
> ---
>
> **[W5] Compare the analysis of the natural Gaussian distribution which is discussed in GUI-G2.**
>
> Thank you for the constructive suggestion. We want to  clarify that GUI-G2 and our method use Gaussian intuition in fundamentally different ways. While, **GUI-G2 uses a Gaussian-shaped reward as a training-time** supervision signal, **ReGUIDE leverages the Gaussian structure **at test time** for KDE-based spatial search. These two approaches are therefore orthogonal in mechanism and purpose.
>
> Moreover, the underlying motivations of Gaussian structure in the two works are fundamentally different. ReGUIDE is motivated by **empirical observation** that the **model’s coordinate likelihoods** naturally form a Gaussian-like distribution around its initial prediction (Figure 3). In contrast, GUI-G2 is motivated by **human clicking behavior** that naturally forms Gaussian distributions centered on target elements and trains the model to mimic human behavior.
>
> Nonetheless, these orthogonal perspectives converge to similar gaussian intuition for grounding systems. This suggests a potential underlying connection between emergent model behavior and human spatial patterns.
>
> Thanks to the author's suggestion, we could add this constructive discussion about comparison between intuition from ReGUIDE and GUI-G2 in revised manuscript.
>
> [8] Tang et al., GUI-G2: Gaussian Reward Modeling for GUI Grounding, preprint 07.2025
>
> ---
>
> **Concluding Remarks**
>
> We believe our additional experiments and clarifications regarding (i) orthogonal adaptation, (ii) relation between components, and (iii) comparing intuitions with other works, will strengthen the paper significantly. Your insightful guidance has been instrumental in refining our analysis. Should you have any inquiries or require clarifications about our rebuttal, please don't hesitate to reach out. We are eager to address any concerns and elucidate potential ambiguities in greater depth.

---

> ### Author Response · Authors · 2025-11-27
> **Further Discussion Before the Deadline**
>
> Dear Reviewer 6X4C,
>
> Thank you once again for your time and thoughtful efforts in reviewing our paper.
>
> As the discussion period will conclude soon, we would like to gently remind you in case you have any remaining comments. We believe that we have sincerely and successfully addressed your concerns, supported by the corresponding additional experimental results.
>
> If you have any further concerns or questions, please feel free to let us know.
>
> Thank you very much, Authors

---

### Official Review · Reviewer_KLeA · 2025-11-01

**Soundness:** 3
**Presentation:** 4
**Contribution:** 3
**Rating:** 6
**Confidence:** 3

**Summary:**

This paper presents ReGUIDE, a data-efficient framework for GUI coordinate grounding that enables multimodal language models to accurately localize interface elements. ReGUIDE employs a two-stage training approach: (1) self-generating reasoning through online reinforcement learning using grounding accuracy as reward, and (2) enforcing spatial consistency between global and local image views using transformation-based priors. At inference, a test-time search strategy combines spatial cropping with KDE-based coordinate aggregation. Using only 20K samples (≈0.2% of UGround), ReGUIDE outperforms prior models, improving Qwen-2.5-VL-3B accuracy from 55.5% to 88.0% on ScreenSpot and from 23.9% to 44.5% on ScreenSpot-Pro, demonstrating significant data efficiency gains.

**Strengths:**

**1. Novel Integrated Data-Efficient Framework:**

The paper proposes a highly novel framework, *ReGUIDE*, that tackles the GUI grounding problem with exceptional data efficiency. Its originality lies in the synergistic integration of components across both training and inference: reinforcement learning for self-generated language reasoning, a subsequent training stage enforcing spatial consistency under transformations, and a final test-time spatial search with KDE-based aggregation. This comprehensive combination effectively addresses the spatial reasoning and localization precision limitations of prior approaches.

**2. Strong and Data-Efficient Performance:**

ReGUIDE achieves significant performance improvements, setting a new competitive standard across multiple challenging GUI grounding benchmarks, including ScreenSpot and ScreenSpot-Pro. Notably, it demonstrates remarkable data efficiency, reaching state-of-the-art results while using only a small fraction of the data (a 20K subset, or 0.2% of samples) compared to prior methods trained on millions of examples.

**3. Comprehensive Empirical Evaluation and Insights:**

The paper presents extensive and rigorous experiments, including systematic ablation studies that isolate the contribution of each training and inference component. Moreover, it validates practical utility through strong transfer performance on downstream agentic tasks such as Multimodal-Mind2Web and AndroidControl, further supporting the empirical soundness of the proposed approach.

**4. Clear Presentation and Reproducibility:**

The paper is well written and well organized, featuring clear figures and structured technical explanations. It provides valuable insights into the model’s internal mechanisms through attention map visualization and analysis of self-generated reasoning pathways. The inclusion of complete experimental details, hyperparameter settings, and the stated plan to release code enhances the clarity and reproducibility of the work.

**Weaknesses:**

**1. Lack of comparison with other RL-based methods:**
For instance, *UI-AGILE-7B*, which also employs reinforcement learning with only 9K examples, achieves 48.7% accuracy on ScreenSpot-Pro when initialized from Qwen2.5-VL, which is comparable to or even surpasses ReGUIDE’s results under a smaller data regime. It would be helpful to discuss this comparison to clearly highlight the differences between the two approaches.

**2. Inference latency:**
ReGUIDE’s performance gain largely depends on its two-stage test-time spatial search. While effective, this procedure increases inference time by approximately 2.0 to 2.2 times (Table 13), introducing a relatively high latency overhead for real-time or interactive GUI agents. This raises concerns about its practicality in latency-sensitive scenarios where fast responses are critical.

**3. Entanglement of test-time scaling and training effects:**
The main results combine the gains from both the training components (RL + Consistency) and the inference enhancements (test-time scaling), making it difficult to isolate the true contribution of the proposed data-efficient training. Since other baselines are evaluated using single-pass inference, a fair comparison requires reporting ReGUIDE’s performance both with and without test-time scaling to clarify how much of the impro

**Questions:**

**1. On the SOTA claim and leaderboard comparison:**

The paper reports state-of-the-art results on ScreenSpot-Pro. However, several stronger 7B baselines have already been listed on the public leaderboard (https://gui-agent.github.io/grounding-leaderboard/), such as GUI-ARP-7B (60.8%), Holo1.5-7B (57.9%), GTA1-7B (55.5%), and GUI-Cursor-7B (56.5%).

It would be helpful if the authors could clarify how ReGUIDE compares with these concurrent approaches and discuss their methodological differences to better position the contribution of this work.

**2. On inference latency and deployability:**

The two-stage test-time search appears to increase inference time by about twofold. Could the authors share their perspective on how this additional latency might affect real-world deployment, particularly for GUI agents requiring real-time interaction?

**3. On isolating training and inference gains:**

The core contribution is argued to be Data Efficient Grounding achieved through the proposed training paradigm. However, the reported SOTA results include the significant gains from the Test-Time Scaling. To ensure a transparent analysis of data efficiency and computational trade-offs, could the authors please report ReGUIDE's performance in the main results before and after applying the Test-Time Scaling? This separation would clearly isolate the performance improvement attributable solely to the data-efficient training methods.

**4. On handling high-resolution inputs:**

The Cropping stage contributes notably to the gains on the high-resolution ScreenSpot-Pro benchmark, suggesting that the base model may still struggle with large input resolutions. Could the authors elaborate on whether the Consistency training helps mitigate this issue by learning a more robust, scale-invariant representation internally, or whether the improvement mainly depends on the external inference-time Cropping process?

---

> ### Author Response · Authors · 2025-11-22
> **Response to Reviewer KLeA (1/3)**
>
> Dear reviewer KLeA,
>
> We sincerely appreciate your valuable comments. We respond to your questions below.
>
> ---
>
> **[W1] Lack of comparison with RL-methods, UI-AGILE also employs RL with smaller data regimes.**
>
> We carefully remark that we already discussed and compared several RL-trained models (e.g., UI-R1, GUI-R1) with ReGUIDE. Nonetheless, we agree that including discussion and comparison about such concurrent works (UI-AGILE) will make the paper clearer.
>
> We clarify that ReGUIDE introduces unique components which are distinct from concurrent RL-trained approaches such as UI-AGILE. Conceptually, UI-AGILE mainly focuses on sophisticated reward-shaping, and relies on test-time verification of general VLM to achieve high score. They do not explicitly exploit model spatial priors and **rely on an external model** at test-time. In contrast, ReGUIDE (trained on 20k random samples) uniquely incorporates.
> (i) consistency learning which exploits aligning spatial prior and
> (ii) KDE-based spatial search which utilizes the natural gaussian property which **doesn't rely on external VLM**.
>
> Regarding direct numerical comparison, we argue that comparing raw numbers requires care due to heterogeneous settings. Specifically, UI-AGILE’s reported performance benefits from external model verification. When, fairly evaluating performance of UI-AGILE-7B with no external model on ScreenSpot-pro **drops to 44.0%** which is **lower than ReGUIDE’s 47.1%**. This confirms that ReGUIDE shows better grounding capabilities within the model itself in a data-efficient setting.
>
> Complementary, to address this more constructively, we focused on verifying whether ReGUIDE’s unique components are complementary to these baselines. To this end, we continually fine-tune publicly available checkpoints of concurrent approaches ( UI-AGILE-7B, GTA1-7B, Holo1.5-7B) using ReGUIDE’s components, namely consistency learning (with only 2k additional samples) and our KDE-based test-time scaling. As shown in the table below, ReGUIDE consistently improves the performance of all base models (e.g., +6.8%p for UI-AGILE-7B), proving that our method captures spatial logic that standard RL misses. These results highlight the unique components of ReGUIDE are model-agnostic and complementary, and can reliably enhance existing GUI grounding methods.
>
>
> \begin{array}{lccc}
> \hline
> \text{Model} & \text{ScreenSpot} & \text{ScreenSpot-v2} & \text{ScreenSpot-Pro} \newline \hline
> \text{GTA1-7B} & - & 92.0 & 50.7  \newline
> \text{  + ReGUIDE} & - & \textbf{92.8} & \textbf{53.4} \newline \hline
> \text{UI-AGILE-7B} & 90.6 & 92.0 & 44.0 \newline
> \text{  + ReGUIDE} & \textbf{92.0} & \textbf{92.8} & \textbf{50.8} \newline \hline
> \text{Holo1.5-7B}  & 91.7 & 93.9 & 57.8 \newline
> \text{ + ReGUIDE}  & \textbf{92.0} & \textbf{94.3} & \textbf{63.2} \newline \hline
> \end{array}
>
> We thank the reviewer for a valuable question and have incorporated these results in the revised manuscript (highlighted in blue).
>
> [1] Lian et al., Ui-agile: Advancing gui agents with effective reinforcement learning and precise inference-time grounding, preprinted arxiv2507\
> [2] Hcompany, Holo1.5 - Open Foundation Models for Computer Use Agents, huggingface\
> [3] Yang et al., GTA1: GUI Test-time Scaling Agent, preprinted arxiv2507
>
> ---
>
> **[Q1] On the SOTA claim, and lack of comparison with concurrent approaches  (e.g., UI-AGILE-7B,GUI-ARP-7B, Holo1.5-7B , GTA1-7B, and GUI-Cursor-7B)**
>
>
> As we clarified in the **response of [W1]**, compared to other baselines, ReGUIDE uniquely exploits spatial prior by introducing (i) consistency learning for aligning spatial prior and (ii) KDE-based spatial search utilizing the inherent gaussian property of VLM. However, we respectfully point out that direct comparison requires care due to heterogeneous training datasets. For example, GTA-1 trained on 100k samples aggregated from 5 different sources, and UI-AGILE uses 9k samples from 4 different sources, which are filtered with external models. However, ReGUIDE relies on 20k randomly sampled data points from the standalone UGround dataset.
>
> Because of this dataset disparity, simply comparing raw numbers may overlook the methodological contributions. Therefore, we believe the fairest way to demonstrate how ReGUIDE orthogonally adapts to other baselines and pushes the SOTA performance.
>
> As shown in **the experiment of [W1]**, ReGUIDE is a model-agnostic framework that can further enhance their grounding capabilities. By adapting ReGUIDE’s component to Holo1.5-7B, ReGUIDE boosts the performance of **Holo1.5-7B (57.8 -> 63.2) on ScreenSpot-Pro**. This result achieves a **new state-of-the-art** among 7B models on the leaderboard.
>
> This result demonstrates that while strong baselines exist, ReGUIDE can serve as a unique, complementary component to push the performance even further.

---

> ### Author Response · Authors · 2025-11-22
> **Response to Reviewer KLeA (2/3)**
>
> **[W2, Q2] On inference latency and deployability**
>
>
> Thank you for pointing this out. It is true that test-time scaling methods naturally incur additional inference latency to gain higher accuracy. However, we argue that this trade-off is (i) practically justified and (ii) fully controllable in real-world deployment.
>
> In real-world usage of GUI-agent, the cost of a false positive is significantly higher than a slight inference delay. Some actions are critical (e.g., purchasing) and hard to recover (e.g., sending e-mail) where recovering from errors often takes more time. Thus, investing in ReGUIDE’s spatial search to ensure high precision reduces the total task completion effort and running time.
>
> Furthermore, ReGUIDE is designed so that latency accuracy trade-off is adjustable by users. Users can control the number of sampled crops N in our KDE-based test-time scaling to match their latency budget. As shown in table below, moving from single-pass inference to the full setting N=16 roughly doubles the inference time while achieving the best accuracy, but a smaller N already offers a favorable balance. For example, with N=4 on ScreenSpot-Pro, the performance retains about 90% of the full setting, while the time cost is only x1.6.
>
> We believe that addressing the high cost of false-positives and offering latency flexibility make ReGUIDE highly practical for real-world scenarios.
>
> \begin{array}{lcc|ccc|ccc}
> \hline
> &&& &ScreenSpot&&&ScreenSpot-pro&\newline \hline
> \text{Method} & \text{Scaling} & N &
> \text{Time (s)} & \text{Time Cost} & \text{Accuracy} &
> \text{Time (s)} & \text{Time Cost} & \text{Accuracy} \newline
> \hline
> & X & - &1.69 & \text{x1.0} & 84.9 &2.43 & \text{x1.0} & 27.9 \newline
> \text{ReGUIDE-3B} & O & 4 &3.06 & \text{x1.8} & 86.9 &3.98 & \text{x1.6} & 41.1 \newline
>  & O & 8 &3.31 & \text{x2.0} & 87.3 &4.29 & \text{x1.8} & 43.5 \newline
>  & O & 16 &3.76 & \text{x2.2} & 88.0 &4.88 & \text{x2.0} & 44.5 \newline \hline
> \end{array}
>
>
> ---
>
> **[W3, Q3] On isolating training and inference gains.**
>
> We would like to clarify that the KDE-based Test-time scaling (TTS) is an integral part of our proposed contribution, not just an external post-processing step. We designed ReGUIDE as a unified system where the training stage and the inference stage work together. Therefore, we believe the full performance represents the true capability of our method.
>
> Nevertheless, we agree that highlighting the training gain is important for clearness. We want to clarify that, in the ablation study (in Table 5 and Table 7) of the original manuscript, we reported each component’s performance gain, which separates the effects of RL reasoning, consistency training, and KDE-based test-time scaling. However, following your suggestion to make this clearer in the main comparison, we have now explicitly separated the performance of ReGUIDE with and without Test-Time Scaling in Table 1.
>
> As shown in the table below, on the ScreenSpot-Pro benchmark with the 7B model, ReGUIDE achieves a **+7.3%p (29.0 -> 36.3) improvement solely from the training components**, and an additional **+10.8%p (36.3 -> 47.1) gain from applying test-time scaling**. These results show that even without test-time scaling, ReGUIDE’s training components yield substantial improvements over SFT baselines under identical data settings, then test-time scaling provides a further boost.
>
> \begin{array}{cc|ccc} \hline
> Model &  \text{KDE-TTS}& \text{ScreenSpot} & \text{ScreenSpot-v2} & \text{ScreenSpot-pro} \\ \newline\hline
> Qwen2.5-VL-7B&X&84.7  & 82.6 & 29.0  \newline
> ReGUIDE-7B&X&88.1 & 91.0& 36.3  \newline
> ReGUIDE-7B&O& \textbf{90.2} & \textbf{92.3} & \textbf{47.1}   \newline
> \hline
> \end{array}
>
> ---

---

> ### Author Response · Authors · 2025-11-24
> **Response to Reviewer KLeA (3/3)**
>
> **[Q4] On handling high-resolution inputs**
>
> Thank you for the insightful question. We clarify that consistency training improves the model’s internal ability to handle scale changes, which makes the cropping much more effective. While cropping allows the models to “see” detailed views, consistency learning teaches the model to “understand” these views correctly. Therefore, the improvement does not come from cropping alone, but from the synergy between the learned internal robustness and the external search.
>
> This synergy is quantitatively shown in our ablation study (Table 5). To isolate the effect of internal representation, we compared the performance of Test-Time Scaling (TTS) with and without Consistency Training. As shown in the summarized table below, the model trained **without consistency achieves 40.7% (+13.5%p)** on ScreenSpot-Pro, whereas the **consistency-trained model reaches 44.5%(+17.3%p)**. This means that consistency learning increases the effectiveness of test-time scaling by **approximately 28% (17.3 vs 13.5)**. This substantial difference indicates that the internal representation learned through consistency learning makes the test-time spatial search significantly more effective.
>
> We believe this behavior is consistent with prior consistency-based robustness methods [4,5,6], which report that enforcing across augmented views stabilizes the representation under input perturbations. In our setting, the global screen and its local crops can be viewed as spatial augmentations. We conjecture that Consistency Learning encourages scale-robust spatial features, allowing the cropping-based search to exploit them more effectively.
>
> \begin{array}{ccc|c}\hline
> \text{RL} & \text{consistency} & \text{TTS} & \text{ScreenSpot-pro} \newline \hline
> O&X& X & 27.2 \newline
> O&X& O & 40.7 \newline
> O&O& O & 44.5 \newline
> \hline
> \end{array}
>
> [4] Hendricks et al., AugMix: A Simple Data Processing Method to Improve Robustness and Uncertainty, ICLR 2020\
> [5] Zhang et al., MEMO: Test Time Robustness via Adaptation and Augmentation, Neurips 2022\
> [6] Xie et al., Unsupervised Data Augmentation for Consistency Training, Neurips 2020
>
> ---
>
> **Concluding Remarks**
>
> We believe our additional experiments and clarifications regarding (i) orthogonal adaptation, (ii) latency-performance tradeoff, and (iii) the relation between training and inference components, will strengthen the contribution of ReGUIDE and improve the clarity of the revised manuscript. We appreciate the reviewer’s constructive feedback, which has been valuable in refining the work.

---

> ### Author Response · Authors · 2025-11-27
> **Further Discussion Before the Deadline**
>
> Dear Reviewer KLeA,
>
> Thank you once again for your time and thoughtful efforts in reviewing our paper.
>
> As the discussion period will conclude soon, we would like to gently remind you in case you have any remaining comments. We believe that we have sincerely and successfully addressed your concerns, supported by the corresponding additional experimental results.
>
> If you have any further concerns or questions, please feel free to let us know.
>
> Thank you very much,
> Authors

---

### Official Review · Reviewer_CEeH · 2025-11-05

**Soundness:** 2
**Presentation:** 3
**Contribution:** 2
**Rating:** 4
**Confidence:** 4

**Summary:**

This paper addresses the critical bottleneck of data inefficiency in training MLLMs for GUI grounding. The authors propose ReGUIDE, a novel framework that achieves state-of-the-art results using a very small fraction--0.2% of the data required by existing baselines. The method's novelty lies in its two-stage training process, which first uses online reinforcement learning to reward accurate coordinate prediction, compelling the model to self-generate its own language-based reasoning without explicit supervision. This is followed by another stage that enforces spatial consistency between global and cropped views of the image. The framework is further enhanced by an effective test-time scaling strategy that uses spatial search and Kernel Density Estimation to iteratively refine coordinate predictions. Experiments across multiple benchmarks, including high-resolution and agentic tasks, demonstrate significant performance gains, highlighting the method's effectiveness in learning robust spatial priors from minimal data.

**Strengths:**

- Data Efficiency. This is the paper's primary contribution and it is highly significant. Achieving nice performance while training on only 20k samples versus 10M is a major step forward for the field, making high-performance grounding more accessible.
- Effective Inference-Time Search. The test-time scaling strategy, which uses KDE for an initial vote, crop, and then vote again, is well-motivated and empirically powerful. The paper shows this is particularly effective for high-resolution images.

**Weaknesses:**

- Two very relevant work, Aria-UI (for GUI grounding SFT) and GTA-1 (for GUI RL training with GRPO) is not discussed nor compared in the paper.
- From Tab.5, the proposed two test-time scaling strategies play important role in model's performance with SS and SS-pro. The questions here would be:
1) since we may easily move the two strategies to existing models like UGround, will they benefit from it?
2) since without the two strategies, the proposed model generally performs on par with the baselines, and considering the authors only used 0.2% of the data, can scaling law in terms data still apply to ReGUIDE models? For example, if using 10x or 100x of the data, how would ReGUIDE benefit from it?

**Questions:**

Please see weaknesses.

---

> ### Author Response · Authors · 2025-11-22
> **Response to Reviewer CEeH (1/2)**
>
> Dear reviewer CEeH,
>
> We sincerely appreciate your valuable comments. We respond to your questions below.
>
> ---
>
> **[W1] Relevant Works (e.g. Aria-UI: SFT-trained and GTA-1: GRPO-trained) are not discussed nor compared in the paper.**
>
> We carefully remark that we already discussed and compared several SFT-trained (e.g., UGround, AGUVIS, SeeClick) and GRPO-trained models (e.g., UI-R1, GUI-R1) with ReGUIDE. Nonetheless, we agree that including discussion and comparison about such concurrent works (Aria-UI, GTA-1) will make the paper clearer.
>
> Conceptually, these concurrent approaches (Aria-UI and GTA-1) focus on other aspects and are complementary to ReGUIDE. Aria-UI relies on a massive GUI dataset (1M samples), and GTA-1 is a GRPO-based RL approach with extensive data-filtering (100k samples). They do not explicitly exploit model spatial uncertainty or geometric priors. In contrast, ReGUIDE is trained on 20k randomly sampled dataset and uniquely introduces (i) consistency learning aligns global-local spatial prior and (ii) KDE-based spatial search which leverages the natural gaussian structure of the VLM’s coordinate likelihood.
>
> Regarding direct numerical comparison, since concurrent works are trained with heterogeneous dataset, we believe that raw numbers should be interpreted with care. To address this issue more constructively, we focus on verifying whether ReGUIDE’s unique components are complementary to concurrent baselines. To this end, we continually fine-tune GTA1-7B which is the strongest among the baselines cited by the reviewer. By applying ReGUIDE’s components to the official GTA-1 checkpoint, we achieved a performance gain of 2.7%p (50.7 -> 53.4) on ScreenSpot-pro. This result confirms that ReGUIDE captures spatial dependencies that GTA-1's standard RL training might miss.
>
> Furthermore, to further verify that this improvement is model-agnostic and not limited to specific models GTA-1, we extended our evaluation to other concurrent baselines like UI-AGILE-7B and Holo1.5-7B. As shown in the table below, **adapted Holo1.5-7B achieved 63.2 on ScreenSpot-pro, a state-of-the-art performance** among open-sourced 7B-scale models. These results highlight the unique components of ReGUIDE are model-agnostic and complementary, and can reliably enhance existing GUI grounding methods
>
> \begin{array}{lccc}
> \hline
> \text{Model} & \text{ScreenSpot} & \text{ScreenSpot-v2} & \text{ScreenSpot-Pro} \newline \hline
> \text{Aria-UI} & 82.4 & - & 11.3  \newline \hline
> \text{GTA1-7B} & - & 92.0 & 50.7  \newline
> \text{  + ReGUIDE} & - & \textbf{92.8} & \textbf{53.4} \newline \hline
> \text{UI-AGILE-7B} & 90.6 & 92.0 & 44.0 \newline
> \text{  + ReGUIDE} & \textbf{92.0} & \textbf{92.8} & \textbf{50.8} \newline \hline
> \text{Holo1.5-7B}  & 91.7 & 93.9 & 57.8 \newline
> \text{ + ReGUIDE}  & \textbf{92.0} & \textbf{94.3} & \textbf{63.2} \newline  \hline
> \end{array}
>
>
> We thank the reviewer for a valuable question and have incorporated these results in the revised manuscript (highlighted in blue).
>
> [1] Lian et al., Ui-agile: Advancing gui agents with effective reinforcement learning and precise inference-time grounding, preprinted arxiv2507\
> [2] Hcompany, Holo1.5 - Open Foundation Models for Computer Use Agents, huggingface\
> [3] Yang et al., GTA1: GUI Test-time Scaling Agent, preprinted arxiv2507
>
> ---

---

> ### Author Response · Authors · 2025-11-22
> **Response to Reviewer CEeH (2/2)**
>
> **[W2-1] Does ReGUIDE’s test-time scaling strategy work in other existing models like UGround.**
>
> We want to clarify that we already reported the individual contribution of each component including KDE test-time scaling component in ablation study Table 5 in our original draft. The results show each component plays an important role and even synergize themselves. For example, training component improve +29.0%p (55.9 -> 84.9) in ScreenSpot, and 4.0%p (23.9->27.9) ScreenSpot-Pro, test-time scaling component improve +3.1%p (84.9-> 88.0) in ScreenSpot, and 16.6%p (27.9 -> 44.5) in ScreenSpot-Pro. The results indicate that every component is playing an important role.
>
> However, following the reviewer’s suggestion, we additionally apply our test-time scaling to UGround. As shown in table below, applying the KDE test-time scaling (KDE-TTS) strategies to these models yields significant performance improvements. For example, **31.1 to 44.3 for UGround in ScreenSpot-pro**. The results suggest that the KDE test-time scaling is model-agnostic and works well across various methods.
>
> \begin{array}{lccc}\hline
> \text{Model} & \text{KDE-TTS} & \text{ScreenSpot-v2} & \text{ScreenSpot-Pro} \newline \hline
> \text{UGround-7B} & X & 89.1 & 31.1 \newline
> \text{UGround-7B} & O & \textbf{90.9} & \textbf{44.3} \newline \hline
> \end{array}
>
> [4] Gou et al., Navigating the Digital World as Humans Do: Universal Visual Grounding for GUI Agents, ICLR2025
>
>
> ---
>
>
> **[W2-2] Can scaling law in terms of data still apply to ReGUIDE models?**
>
> Due to time constraints, we were unable to experiment with scaling to 10x or 100x. However, we conducted experiments with a smaller dataset size trained from Qwen-2.5-VL-3B, scaling it by up to **2.5x** (from 20k to 50k). As shown in the table below, ReGUIDE continues to benefit from additional data, though with diminishing returns. For example, ScreenSpot-pro increased from **44.5 to 46.0** as data samples increased from 20k to 50k. We believe these trends indicate that a scaling-law behavior does hold for ReGUIDE.
>
> \begin{array}{ccc}\hline
> \text{Train Size} & \text{ScreenSpot} & \text{ScreenSpot-Pro} \newline \hline
> 3.2\text{k} & 83.2 & 29.0 \newline
> 6.4\text{k} & 86.6 & 38.3 \newline
> 9.5\text{k} & 86.5 & 41.4 \newline
> 12.8\text{k} & 86.9 & 42.4 \newline
> 16\text{k} & 87.7 & 43.6 \newline
> 20\text{k} & \textbf{88.0} & 44.5 \newline
> 50\text{k} & \textbf{88.0} & \textbf{46.0} \newline \hline
> \end{array}
>
> ---
>
> **Concluding Remarks**
>
> We believe our additional experiments and clarifications regarding (i) orthogonal adaptation, (ii) the effectiveness of test-time scaling, and (iii) the scaling-law behavior, will strengthen the paper significantly. Your insightful guidance has been instrumental in refining our analysis and improving the clarity of the revised manuscript. We are happy to clarify any remaining questions during the discussion.

---

> ### Author Response · Authors · 2025-11-27
> **Further Discussion Before the Deadline**
>
> Dear Reviewer CEeH,
>
> Thank you once again for your time and thoughtful efforts in reviewing our paper.
>
> As the discussion period will conclude soon, we would like to gently remind you in case you have any remaining comments. We believe that we have sincerely and successfully addressed your concerns, supported by the corresponding additional experimental results.
>
> If you have any further concerns or questions, please feel free to let us know.
>
> Thank you very much,
> Authors

---

### Author Response · Authors · 2025-11-22
**Common Response**

Dear reviewers and AC,

We sincerely appreciate your valuable time and effort spent reviewing our manuscript.

As reviewers highlighted, we believe that our paper provides a data-efficient **(all reviewers)** and effective (**all reviewers**) framework for GUI grounding by integrating novel data-efficient learning (**CEeH, KLeA**) and effective KDE-based test-time search(**all reviewers**), validated through comprehensive evaluations (**all reviewers**) followed by a clear presentation (**all reviewers**).

We appreciate for your constructive comments. In response, we have carefully revised and enhanced the manuscript with the following additional discussions and experiments:

- Orthogonal adaptation of ReGUIDE (**new SOTA**) and compared with extended baselines (Table 2)
- Discussion of the orthogonal adaptation of ReGUIDE and extended baselines (Section 2, Section 4.1)
- Comparison of the gaussian distributed intuition with GUI-G2 (Section 4.2)
- Clarity of offline-agent evaluation results (Table 8)
- Experiments of applying KDE test-time scaling on UGround (Appendix B.5, Table14)
- Extended experiments on the computation cost of inference (Appendix B.6, Table 15)

In the revised manuscript, these updates are temporarily highlighted in “Blue” for your convenience to check.

We hope our response and revision sincerely address all the reviewers’ concerns.

Thank you very much,\
Authors.

---

### Author Response · Authors · 2025-12-03
**AC Letter**

Dear Area Chairs, Senior Area Chairs, and Program Chairs,

Following our previous revision summary, we provide this concise follow-up describing how each reviewer's individual concerns have been addressed. The goal is to make verification straightforward for the decision committee.

---

**Reviewer CEeH**

W1. Relevant works (SFT-trained, RL-trained)  are not discussed and compared.
- Clarified that several SFT-trained, RL-trained relevant works were already discussed and compared.
- Emphasized the conceptual difference that ReGUIDE uniquely exploits spatial priors via consistency learning and KDE-based spatial search.
- Added new experiments where ReGUIDE’s components are orthogonally applied to relevant works, showing consistent gains and leading to a new 7B-SOTA.

W2-1. Does ReGUIDE test-time scaling can adapt to other baselines.
- Conducted an experiment applying ReGUIDE’s KDE-based test-time scaling to other baselines, showing consistent performance improvements.

W2-2. Scaling-Law of ReGUIDE.
- Conducted a controlled data-scaling experiment (increasing training data from 20k to 50k), showing the result consistent with a scaling-law trend.

---

**Reviewer KLeA**

W1/Q1. SOTA claim and comparison with concurrent RL-based approaches.
- Clarified that the primary goal is to propose a data-efficient framework, which evaluated under controlled experiments (same model and dataset), rather than to compete on raw leaderboard scores under heterogeneous data and settings.
- Added experiments where ReGUIDE’s components are orthogonally applied to relevant works, showing consistent gains and achieving a new 7B-level SOTA.

W2/Q2. Inference latency and deployability.
- Discussed that in GUI-agent scenarios, accuracy and reliability are often more critical than minimal latency.
- Highlighted that users can explicitly control the latency–accuracy trade-off by adjusting the number of test-time samples in KDE-based scaling.
- Expanded a latency-accuracy analysis for different sampling budgets to quantify trade-off

W3/Q3. Isolating training, KDE-based test-time scaling gains.
- Clarified that we already analysed isolated gain from each component in the original draft.
- Added results without KDE-based test-time scaling in the main table

Q4. Handling high-resolution inputs and the role of consistency learning.
- Emphasized an analysis comparing KDE-based TTS with and without consistency learning, showing that consistency significantly amplifies the effectiveness of spatial search.
- Discussed and related this to consistency-based robustness studies and framed our interpretation

---


**Reviewer 6X4C**

W1/W2. Comparison with additional concurrent baselines.
- Clarified that several concurrent baselines were already discussed in the original draft, and added newly mentioned baselines.
- Emphasized the conceptual difference that ReGUIDE uniquely exploits spatial priors via consistency learning and KDE-based spatial search.
- Added new experiments where ReGUIDE’s components are orthogonally applied to relevant works, showing consistent gains and leading to a new 7B-SOTA.

W3. Clarity of a table’s contents
- Fixed column naming and revised the caption and text.

W4. Relationship between consistency fine-tuning and attention-grounding.
- Emphasized an analysis comparing KDE-based TTS with and without consistency learning, showing that consistency significantly amplifies the effectiveness of spatial search.
- Discussed and related this to consistency-based robustness studies and framed our interpretation

W5. Comparison with GUI-G2’s Gaussian analysis.
- Clarified that GUI-G2 used a Gaussian reward in training-time reward design, while ReGUIDE uses an empirically observed Gaussian likelihood at test time for KDE, making the two approaches orthogonal.
- Clarified that both works arrive at a similar Gaussian intuition from different angles.

---

We sincerely thank the reviewers, AC, SAC, and PCs for their careful evaluation. We believe that our revised manuscript will be a valuable addition to ICLR 2026.

Best regards,
Authors

---

### Meta-Review · Area_Chair_RPNM · 2026-01-11

**Summary:**

The reviewers generally appreciated the paper's focus on data-efficient GUI grounding and the proposed ReGUIDE framework. However, the reviewers brought up several critical concerns shared across the reviewer panel:

1. Missing Baselines and State-of-the-Art (SOTA) Validity: A primary concern raised by all reviewers (CEeH, KLeA, 6X4C) was the omission of critical concurrent and relevant works, specifically GTA-1, Aria-UI, UI-AGILE, and GUI-G2. The reviewers felt that without these comparisons, the paper's claims of achieving SOTA performance were unsubstantiated or potentially misleading.

2. Inference Latency and Practicality: Reviewer KLeA highlighted that the reliance on a two-stage test-time spatial search (KDE-based cropping and voting) approximately doubles the inference latency. This raises significant concerns regarding the deployment of the model in real-time or interactive agentic workflows where latency is a bottleneck.

3. Attribution of Gains: Reviewers (KLeA, 6X4C) were concerned about the entanglement of gains derived from the training method (RL + consistency) versus the test-time compute (scaling). There was skepticism regarding whether the core training contribution alone was sufficient to outperform strong baselines.

4. Novelty and Overlap: Reviewer 6X4C noted that the analysis of the natural Gaussian distribution of coordinate tokens appeared very similar to findings in GUI-G2, questioning the novelty of this specific insight.

**Reviewer Concerns:**

The authors made good effort to address the missing baselines by running orthogonal adaptation experiments. They demonstrated that ReGUIDE's components (consistency training and KDE search) could be applied to models like GTA-1 and Holo1.5 to boost their performance. The authors provided data isolating the training vs. inference gains, showing that while test-time scaling provides a large boost, the training method does offer improvements over standard SFT. The authors corrected the errors regarding the Android Control benchmarks pointed out by Reviewer 6X4C.

Issues remaining:

Benchmarking Fairness: While the authors showed that their components are complementary to strong baselines (like GTA-1), the reviewers were not fully convinced that ReGUIDE as a standalone system outperforms these state-of-the-art models when comparing.

Latency Trade-off: The concern regarding the 2x latency increase remains a practical barrier. While the authors argued that accuracy is paramount, for an interactive agent, doubling the inference time is a significant cost that the current performance gains may not fully justify compared to faster, single-pass baselines.

Incremental Novelty: The distinction between ReGUIDE's "test-time" Gaussian usage and GUI-G2's "reward-based" Gaussian usage, while technically valid, did not sufficiently assuage Reviewer 6X4C's concerns about the novelty of the insight.

**Reviewer Scores:**

Reviewer CEeH (Score: 4) likely would have maintained their score of 4. While they acknowledged the rebuttal, they did not update their score, suggesting that the additional experiments did not fundamentally change their view on the paper's positioning relative to the missing related works.

Reviewer KLeA (Score: 6) was the most positive, and might be less so after seeing the extent of the missing baselines highlighted by other reviewers and the confirmation of the high latency costs likely would have tempered their assessment of the paper's "Strong" contribution rating.

Reviewer 6X4C (Score: 4) would almost certainly have kept their score at 4. They were specifically concerned with the similarity to GUI-G2 and the strength of the baselines; the rebuttal clarified the differences but likely did not prove that ReGUIDE represents a substantial enough leap over the existing, omitted literature.

---

### Decision · Program_Chairs · 2026-01-26

Reject